



# Effects of extraction conditions on the redox properties of soil organic matter (SOM) and its ability to stimulate microbial iron(III) mineral reduction by electron shuttling

Yuge Bai[1], Edisson Subdiaga[2], Stefan B. Haderlein[2], Heike Knicker[3], Andreas Kappler[1]

[1]Geomicrobiology, Center of Applied Geosciences, University of Tuebingen, Tuebingen, 72074, Germany
[2]Environmental Mineralogy and Chemistry, Center of Applied Geosciences, University of Tuebingen, Tuebingen, 72074, Germany
[3]Instituto de Recursos Naturales y Agrobiología de Sevilla, Consejo Superior de Investigaciones Científicas (IRNAS-CSIC), Seville, 41012. Spain

*Correspondence to*: Andreas Kappler (andreas.kappler@uni-tuebingen.de)

**Abstract.** Soil organic matter (SOM), including humic substances (HS), is redox-active, can be microbially reduced, and transfers electrons in an abiotic reaction to Fe(III) minerals thus serving as electron shuttle. The standard procedure to extract HS from soil and separate them into humic acids (HA) and fulvic acids (FA) involves alkaline and acidic solutions potentially leading to unwanted changes in SOM chemical and redox properties. To determine the effects of extraction
conditions on the redox and electron shuttling properties of SOM extracts, we prepared HS and SOM extracts from a forest soil applying either a combination of 0.1 M NaOH and 6 M HCl, or water (pH 7). Both chemical extractions (NaOH/HCl) and water extractions were done in separate setups under either oxic or anoxic conditions. Furthermore, we applied the NaOH/HCl treatment to a subsample of the water-extracted-SOM. We found that soil extraction with NaOH lead to ca. 100 times more extracted C and the extracted HS had 2-3 times higher electron exchange capacities (EEC) than SOM extracted
by water. For water-extracted SOM, anoxic extraction conditions lead to about 7 times more extracted C and 1.5 times higher EEC than under oxic extraction conditions. This difference was probably due to the occurrence of microbial reduction and dissolution of Fe(III) minerals in the soil during the water extraction at neutral pH and the concomitant release of Fe(III) mineral-bound organic matter. NaOH/HCl treatment of the water-extracted SOM lead to 2 times higher EEC values in the HA isolated from the SOM compared to the water-extracted SOM itself, suggesting the chemical treatment with NaOH and
HCl caused changes of redox-active functional groups of the extracted organic compounds. Higher EEC of extracts in turn resulted in a higher stimulation of microbial Fe(III) mineral reduction by electron shuttling, i.e. faster initial Fe(III) reduction rates, and in most cases also in higher reduction extents. Our findings suggest that SOM extracted with water at neutral pH should be used to better reflect environmental SOM redox processes in lab experiments and that potential artefacts of the chemical extraction method and anoxic extraction condition need to be considered when evaluating and comparing abiotic
and microbial SOM redox processes.



## 1 Introduction

Soil organic matter (SOM) contains more organic carbon than the sum of the atmosphere and living plants (Fischlin, 2007) and can influence greenhouse gas emission, plant growth and water quality (Lal, 2004; Marin-Spiotta et al., 2014). Studying SOM is challenging because it needs to be separated from other soil components before doing laboratory experiments

(Lehmann and Kleber, 2015). One of the most commonly used methods is a chemical extraction of humic substances (HS) at pH >12 (Achard, 1786). Although the concept of HS as large-molecular-weight molecules formed by degradation and repolymerization of biomolecules has been challenged by seeing SOM as a continuum of progressively decomposing organic compounds (Lehmann and Kleber, 2015), HS extraction is still applied by many laboratories and the extracted HS are still widely used as a proxy for SOM. Briefly, HS are extracted by adjusting the pH to >12 using NaOH, followed by

acidification of the alkaline extract to pH <2 to separate humic acids (HA) from fulvic acids (FA) (Achard, 1786). Ion exchange resins, dialysis, and even hydrofluoric acid (HF) treatment are used to further purify the extracts (IHSS, 2017). Concerns regarding the effectiveness of this harsh chemical extraction method were already raised in 1888 (van Bemmelen, 1888) and last until today (Lehmann and Kleber, 2015; Kleber and Lehmann, 2019).

It has been shown that alkaline extraction influences the chemical composition and the content of redox-active quinoid

moieties of the extracted SOM (Piccolo, 1988; Engebretson and Von Wandruszka, 1999). Participation in redox reactions is a key property of SOM and relevant for many biogeochemical processes in the environment (Murphy et al., 2014). For example, under anoxic conditions, SOM can accept electrons from microorganisms, transfer electrons to other electron acceptors such as Fe(III) minerals, and be reoxidized to accept electrons again from microorganisms (Lovley et al., 1996; Kappler et al., 2004; Bauer and Kappler, 2009; Wolf et al., 2009). This electron shuttling process, which is facilitated by

SOM, can significantly increase microbial reduction rates of poorly soluble Fe(III) minerals (Lovley et al., 1996; Jiang and Kappler, 2008), enable microbial reduction of otherwise inaccessible Fe(III) minerals (Lovley et al., 1998), and stimulate indirect reduction of minerals that are spatially separated from the bacteria (Lies et al., 2005). Highly purified FA and HA are used in most electron shuttling studies to represent SOM (Lovley et al., 1998; Lovley and Blunt-Harris, 1999; Lies et al., 2005; Bauer and Kappler, 2009; Wolf et al., 2009; Klupfel et al., 2014). However, currently it is not known if and to which

extent the SOM electron shuttling capacity is based on protocol-induced changes caused by the harsh chemical isolation procedure.

Piepenbrock and co-authors extracted SOM from a forest soil at circumneutral pH using water (Piepenbrock et al., 2014). Compared to chemically extracted Pahokee Peat humic acids (PPHA), the extracted SOM had a significantly lower reducing capacity (μeq/g C), which was calculated from the concentration of reduced Fe(II) after the abiotic reaction of PPHA/SOM

with Fe(III)-citrate. This potentially indicates different types and proportions of functional groups in these samples. However, due to the different origin of the extracted soils, it remains unclear whether and to which extent the differences in reducing capacities of the SOM extract and PPHA was caused by the chemical extraction methods. Furthermore, in this



study the water extraction was conducted only under oxic conditions. Although it is known that the presence of $O_2$ causes oxidation of certain organic compunds under alkaline conditions and therefore chemical extraction with NaOH should be conducted under anoxic conditions (Bauer and Kappler, 2009; Maurer et al., 2010), it remains unclear whether and how the presence of $O_2$ influences the abundance of different (redox-active) functional groups and therefore the redox activity of the water-extracted organic compounds under neutral pH.

To determine the effect of these chemicals on the SOM redox properties, we extracted OM from a forest soil using several methods (Fig. 1). The first was the traditional chemical extraction method (1 M NaOH followed by 6 M HCl) yielding HA and FA under either oxic or anoxic conditions. The second was SOM extraction by deionized water at neutral pH (water-extracted SOM) under either oxic or anoxic conditions. Additionally, we treated the water-extracted SOM with NaOH and HCl to further separate HA and FA from the water-extracted SOM (also under either oxic or anoxic conditions). We analyzed the electron accepting capacity (EAC, i.e. the number of electrons that can be accepted), the electron donating capacity (EDC, i.e. the number of electrons that can be donated by the SOM) and the electron exchange capacity (the sum of EAC and EDC) of all extracted SOM/FA/HA fractions and performed a microbial Fe(III) mineral reduction experiment with all of the different extracts. The goals of this study were, first, to identify the effects of alkali and oxygen on the EEC values of the extracted SOM/FA/HA samples, and second, to compare the rates and extents of microbial ferrihydrite reduction in the presence of the different extracts.

## 2 Materials and methods

### 2.1 Soil organic matter (SOM) extraction

Top soil (0-15 cm) was collected from the Schönbuch forest, Baden-Wuerttemberg, Germany. The forest is dominated by beech with populations of oak, spruce and bald cypress and the soil is qualified as vertic cambisol (world reference base for soil, WRB) (WRB). Soil was dried (30°C), ground to pass through a 2 mm sieve by automatic soil grinder (ball mill, FRITSCH, Germany) and stored in the dark at 4°C. Chemical extraction of FA/HA was modified from the IHSS protocol (IHSS, 2017) as follows. In the IHSS protocol, soil samples are incubated with 0.1 M HCl with a ratio of 10 mL liquid per g dry soil and after 1 h of extraction and XAD-8 resin separation, a fraction called "FA_extract_1" is collected. In our experiment, we did not extract this fraction to avoid using XAD-8 separation. As shown in Fig. 1, 100 g soil was incubated with 400 mL of 0.1 M NaOH (pH 12) for 4 h (overhead shaker, 90 rpm, 25°C). The slurry was centrifuged (5000 rpm, 30 min) and the supernatant was acidified (pH 2) by 6 M HCl to separate FA (dissolved in the supernatant) and HA (precipitated). Within the present study we define these extracts as chemically-extracted FA and chemically-extracted HA. SOM water extracts were prepared following Piepenbrock et al. (Piepenbrock et al., 2014) (Fig. 1). 100 g of soil were incubated with 400 mL of doubly-deionized water (resistivity=18.2 MΩ.cm, 25°C; Milli-Q, Millipore) at pH 7. The pH was





monitored during the extraction and it ranged between 7.02-7.33. The slurry was centrifuged after 24 h; we define the supernatant as water-extracted SOM. A 200 mL aliquot of the supernatant containing the water-extracted SOM was amended

with 1 M NaOH until pH 12. After 4 h of incubation, the pH of the solution was adjusted to <2 by 6 M HCl to precipitate HA and to separate FA, this FA and HA are defined as water-extracted FA and water-extracted HA. Anoxic chemical extractions (NaOH/HCl) and water extractions were performed in a $N_2$-filled anoxic glovebox; filtration (0.45 μm, polyethersulfone membrane (PES), Millipore, Germany) was used instead of centrifugation to remove remaining soil after the extraction. All collected extracts were freeze-dried under oxic conditions and stored (4°C, dark) until use.

**2.2 Total organic carbon (TOC) quantification**

TOC of SOM and FA extracts was quantified directly from their extracted solutions after dilution, whereas the precipitated HA were first freeze-dried and then re-dissolved in Milli-Q water (pH 7, stirring at 300 rpm overnight). All solutions were analyzed for TOC (TOC analyzer, model 2100S, Analytik Jena, Germany).

**2.3 Nuclear magnetic resonance (NMR) measurements**

NMR analysis was conducted from freeze-dried SOM/HA/FA samples. Solid-state $^{13}$C NMR spectra were obtained with a Bruker Avance III HD 400 MHz Wideboard operating at a frequency of 100.63 MHz using zirconium rotors of 4 mm OD with KEL-F-caps. The cross polarization magic angle spinning (CPMAS) technique was applied during magic-angle spinning of the rotor at 14 kHz. A ramped $^1$H-pulse was used during a contact time of in order to circumvent spin modulation of Hartmann-Hahn conditions. A contact time of 1 ms and a 90° $^1$H-pulse width of 2.2 μs were used for all spectra. The $^{13}$C-

chemical shifts were calibrated to tetramethylsilane (0 ppm) and were calibrated with glycine (176.04 ppm). The aromaticity of samples were calculated following a previous study (Abelmann et al., 2005).

**2.4 Fluorescence spectra and 2-D excitation-emission-matrices (EEMs)**

The preparation of samples was modified from a previous study (Chen et al., 2003b). Freeze-dried SOM/FA/HA powders were dissolved in Milli-Q water (pH 7) at a concentration of 100 mg C $L^{-1}$ and the solutions were agitated for 12 h at 300

rpm at room temperature, samples were then filtered through 0.45 μm syringe filters (mixed cellulose ester (MCE), Millipore, Germany). For fluorescence analysis, samples were prepared by stepwise dilution of extract solutions with Milli-Q water (pH 7) until absorbance values of 0.300 at 254 nm wavelength were reached. 1.8 mL of sample were placed in a Suprasil UV quartz rectangular fluorescence cuvette; Milli-Q water was used as blank. All fluorescence spectra were recorded on a fluorescence spectrophotometer equipped with both excitation and emission monochromators (Fluoromax-4,

Jobin-Yvon-SPEX instruments, New Jersey, USA). A 450 W Xenon arc lamp was used as the excitation source. A series of emission scans were collected over a range of excitation wavelengths to generate the fluorescence maps as 2-D excitation-



emission-matrices (EEM). A wavelength step size of 10 nm was used for the collection of EEM spectra. The excitation and emission wavelength ranges were from 200-600 and 300-700 nm, respectively.

## 2.5 Specific UV absorbance at 254 (SUVA$_{254}$) analysis

SUVA$_{254}$ analysis was conducted from SOM/HA/FA solutions dissolved in Milli-Q water at concentrations of 10 mg C L$^{-1}$. All solutions were filtered with 0.45 mm syringe filter (polyethersulfone membrane (PES), Millipore, Germany) and the dissolved organic carbon (DOC) concentration of all samples was analyzed prior to the SUVA$_{254}$ analysis (DOC analyzer, model 2100S, Analytik Jena, Germany). The SUVA$_{254}$ values of all samples were measured in a 1 cm rectangular quartz cuvette with a fluorescence spectrophotometer (Fluoromax-4, Jobin-Yvon-SPEX instruments, New Jersey, USA). The final

SUVA$_{254}$ values of all extracts were calculated with equation (1):

$$SUVA254 = UV_{254}/ DOC \qquad (1)$$

where UV$_{254}$ is the absorbance at 254 nm and 1 cm optical path length.

## 2.6 Electrochemical analysis

Electrochemical analysis followed the method described by Aeschbacher et al (Aeschbacher et al., 2010). Freeze-dried

extracts (powders) were dissolved in 100 mM of phosphate buffer (pH 7) at a concentration of 100 mg C L$^{-1}$. After overnight agitation at 300 rpm at room temperature, samples were p through 0.22 μm syringe filters (mixed cellulose ester (MCE), Millipore, Germany). All preparations and measurements were conducted in an anoxic glovebox. The number of electrons transferred to and from all extracts were quantified by integration of reductive and oxidative current responses after baseline correction in mediated electrochemical reduction (MER; at E$_h$=-0.49 V) and mediated electrochemical oxidation (MEO;

E$_h$=+0.61 V) with 1'1-ethylene-2,2'-bipyridyldiylium di-bromide (DQ) and 2,2'-azino-bis(3-ethylbenzothiazoline-6-sulphonic acid) (ABTS) as electron transfer mediators, respectively. To obtain the EAC and EDC values, the integrated current response was normalized to the measured DOC of all extracts prior to the EEC analysis (DOC$_{SOM/FA/HA}$ [mg C/L], DOC analyzer, model 2100S, Analytik Jena, Germany) as shown in Eq. 2 and 3:

$$EAC=I_{red}dt/ (F \times DOC_{SOM/FA/HA}) \qquad (2)$$

$$EDC=\int I_{ox}dt/F \times DOC_{SOM/FA/HA} \qquad (3)$$

where I$_{red}$ and I$_{ox}$ ([A]) are baseline-corrected reductive and oxidative currents in MER and MOR, respectively (F=Faraday constant).

## 2.7 Microbial Fe(III) reduction experiment and calculation of microbial Fe(III) reduction rates

Solutions of organic matter extracts for the microbial Fe(III) reduction experiment were prepared by dissolving freeze-dried

powders in 50 mM phosphate buffer (pH 7.0-7.2) at 500 mg C L$^{-1}$, agitation overnight (300 rpm, room temperature),





filtration and sterilization (0.22 μm syringe filters, mixed cellulose ester (MCE), Millipore, Germany), as described before (Jiang and Kappler, 2008). Although the chosen phosphate concentration is higher than typically observed in nature and can potentially lead to the formation of Fe(II) phosphate minerals (e.g., vivianite) during our microbial Fe(III) reduction experiment, this phosphate buffer was chosen to enable comparison of our study to previous studies (Jiang and Kappler,

2008; Bauer and Kappler, 2009; Klupfel et al., 2014; Piepenbrock et al., 2014). All solutions were deoxygenated 3 times (each time 3 min vacuum and 3 min $N_2$-flushing) and stored in dark bottles to avoid photochemical reactions. Ferrihydrite was prepared as described before (Amstaetter et al., 2012) and stored no more than 2 months (4°C) before use.

*Shewanella oneidensis* MR-1 cells from a frozen stock were streaked on oxic lysogeny broth (LB) agar plates (10 g/L peptone, 5 g/L yeast extract, 10 g/L NaCl and 15 g/L agar). Colonies were transferred to liquid LB medium and incubated at

30°C for 14 h, harvested by centrifugation (10 min, 8000 rpm) and then washed three times with anoxic SBM medium (0.225 g/L $K_2HPO_4$, 0.225 g/L $KH_2PO_4$, 0.46 g/L NaCl, 0.225 g/L $(NH_4)_2SO_4$, 2.18 g/L Na-lactate, 0.117 g/L $MgSO_4 \cdot 7H_2O$, 2.38 g/L HEPES, pH 7.2-7.5). For the Fe(III) reduction experiments, washed cells were added at a final concentration of $10^7$ cells $mL^{-1}$ to solutions of SOM/HA/FA (50 mg C $L^{-1}$) and ferrihydrite (15 mM) in SBM medium (phosphate in the SBM medium was 5 mmol $L^{-1}$). The headspace was flushed with $N_2$ and the bottles were incubated in the dark (30°C). At each

sampling point, an 100 μl aliquot was taken from each bottle, acidified and incubated with 900 μl of 1 M HCl for 1 h to facilitate mineral dissolution, centrifuged (14600 rpm, 5 min), and the total Fe(II) concentration was quantified with the spectrophotometric ferrozine assay in a microtiterplate (Stookey, 1970;Hegler et al., 2008). The fastest reduction rates of the microbial Fe(III) reduction experiments were calculated as shown and explained in the supporting information (SI) (Fig. S4).

## 3 Results

### 3.1 Quantity of soil organic carbon extracted by different methods and characterization of extracted SOM, HA and FA

Comparison of different extraction methods revealed that the amount of soil-extracted carbon varied depending on the presence/absence of $O_2$ during the extraction and on the type of extraction liquid (Tab. 1). Extraction with $H_2O$ at neutral pH under oxic conditions followed by NaOH and HCl treatment to separate HA and FA in the SOM extract yielded 0.036, 0.021

and 0.014 g C in the SOM, FA and HA fractions, respectively, corresponding to 0.41, 0.24 and 0.15% of the total carbon present in the soil. In contrast, anoxic water extraction significantly increased the fraction of extracted carbon to 0.234, 0.146, and 0.079 g C in the SOM, FA and HA fractions, respectively, corresponding to 2.74, 1.69 and 0.90% of the total soil C. Chemical extraction using NaOH and HCl under oxic conditions yielded ca. 100 times more carbon for FA (1.451 g; 17.0% of the total carbon present in the soil) and HA (1.450 g; 17.0%) compared to water under oxic conditions at neutral

pH. Under anoxic conditions, the chemical extraction also lead to a somewhat higher percentage of extracted carbon for FA (20.7%) and HA (22.0%) than under oxic conditions.



Specific UV absorbance at 254 nm (SUVA$_{254}$) indicates the aromaticity of the extracted SOM, FA and HA (Tab. 1). SOM extracted with water showed a SUVA$_{254}$ value of 0.018 mg$^{-1}$ C cm$^{-1}$ under oxic conditions and the value increased to 0.027 mg$^{-1}$ C cm$^{-1}$ when extracted under anoxic conditions. Similarly, for both FA chemically isolated from the water-SOM
extracts and FA chemically extracted from soil, the SUVA$_{254}$ values were higher under anoxic conditions than under oxic conditions. All HA extracts showed much higher SUVA$_{254}$ values than SOM and FA extracts, with the highest value of 0.265 mg$^{-1}$ C cm$^{-1}$ (HA chemically-extracted under anoxic conditions), followed by HA chemically isolated from the water-extracted SOM under oxic conditions (0.207 mg$^{-1}$ C cm$^{-1}$). For HA chemically isolated from the water-extracted SOM under anoxic conditions and HA chemically extracted from soil under oxic conditions, the SUVA$_{254}$ values are 0.068 and 0.083
mg$^{-1}$ C cm$^{-1}$, respectively. A 2-way ANOVA statistical analysis revealed that both oxic versus anoxic conditions and the extraction method (neutral pH water versus chemical extraction) resulted in different SUVA$_{254}$ values at a significance level of $P<0.05$ (Tab. S1). In general, anoxic conditions and the chemical extraction method lead to higher SUVA$_{254}$ values of the extracts, suggesting that these extracts had a higher degree of aromaticity (Korshin et al., 1997). $^{13}$C-NMR and EEM analysis of extracted SOM, FA and HA (Tab. S2 and Fig. S1) confirmed e higher contents of aromatic carbon in samples subject to
chemical extraction or anoxic conditions.

Furthermore, after 24 h water extraction of SOM, we found a maximum Fe(II) concentration of 3 mmol L$^{-1}$ in the extracted SOM solution. Although more than 90% of the Fe was removed by filtration (0.22 μm, mixed cellulose ester (MCE), Millipore, Germany) and around 30% of the remaining Fe(II) was oxidized to Fe(III) during the oxic freeze drying process, there was still 15-123 μmol L$^{-1}$ total Fe present in the water-extracted SOM/FA/HA fractions (Tab. S3). This Fe is potentially
redox-active and can contribute to the redox properties of the extracted OM. However, since this Fe is an integral part of the OM in the environment and we were interested in determining the role of environmentally relevant OM extracts in electron shuttling, we decided not to further purify the extracts (also because this probably would have changed the properties of the present redox-active organic matter).

**3.2 Redox properties of extracted SOM, FA and HA**

We used mediated electrochemical reduction and oxidation to quantify the EAC, EDC and thus the EEC of all SOM, FA, and HA extracts (Fig. 2). Based on the Fe content we calculated the contribution of the Fe to the redox properties of all extracts (Tab. S3). A 2-way ANOVA statistical analysis revealed that both the extraction condition (anoxic versus oxic) and the extraction method (neutral pH water versus NaOH) resulted in significantly different EEC values ($P<0.05$; Tab. S4). The
EEC of water-extracted SOM obtained under oxic conditions was 32 μmol e$^{-}$ mmol C$^{-1}$ (with ca. 4 μmol e$^{-}$ mmol C$^{-1}$ from Fe), whereas when extracted anoxically, it increased to 45 μmol e$^{-}$ mmol C$^{-1}$ (with 14.8 μmol e$^{-}$ mmol C$^{-1}$ from Fe). Higher EEC values under anoxic compared to oxic extraction conditions were also observed for all extracted FA: for FA isolated





oxically from the water-extracted SOM (water-extracted FA, oxic), the EEC was 13 µmol e- mmol $C^{-1}$ (2.3 µmol $e^-$ mmol $C^{-1}$ from Fe), while it increased to 24 µmol $e^-$ mmol $C^{-1}$ (2.7 µmol $e^-$ mmol $C^{-1}$ from Fe) when FA was isolated anoxically from

the water-extracted SOM (water-extracted FA, anoxic). The EEC of chemically-extracted FA under anoxic conditions was even 33 µmol $e^-$ mmol $C^{-1}$ higher than FA chemically-extracted under oxic conditions. Similar to FA, for the HA isolated from water-extracted SOM (water-extracted HA), the EEC values increased from 15 µmol $e^-$ mmol $C^{-1}$ (1.9 µmol e- mmol $C^{-1}$ from Fe) under oxic conditions to 83 µmol $e^-$ mmol $C^{-1}$ (7.3 µmol e- mmol $C^{-1}$ from Fe) under anoxic conditions. For chemically-extracted HA, EEC values increased from 40 µmol e- mmol $C^{-1}$ under oxic conditions to 127 µmol e- mmol $C^{-1}$

under anoxic conditions.

The total number of electrons that can be exchanged (that means transferred from Fe(III)-reducing bacteria to the OM, or from the OM to Fe(III) minerals) by water-extracted SOM before and after the chemical separation of FA and HA from this SOM was also calculated (the recovery of EEC) under both oxic and anoxic conditions (Fig. 2). For the extracts obtained under anoxic conditions, the sum of total exchangeable electrons values of the water-extracted FA and HA (786 µmol e-) was

almost identical to that of water-extracted SOM before NaOH/HCl treatment (836 µmol e-). In contrast, under oxic conditions, the sum of the EEC values of the FA and HA separated from the water-extracted SOM was 324 µmol e-, ca. 5-times higher than the EEC value of the water-extracted SOM (64 µmol e-). This confirms that the traditional chemical extraction procedure conducted under oxic conditions stongly enhances the redox capacity of the samples.

In addition to the EEC that represents the total amount of electrons that can be stored by the extracted organic compounds,

the individual EAC and EDC values can be used to characterize the redox state of the SOM, HA and FA. The EDC and EAC quantify how many electrons are already stored in the molecules (EDC) and how many electrons can still be taken up by functional groups that can be reduced (EAC) (Fig. 2). Surprisingly, the EAC values were larger for all FA and HA extracts obtained under anoxic extraction conditions than under oxic conditions (Fig. 2). The higher EAC under anoxic conditions suggests the presence of more functional groups that can be reduced in FA and HA extracted in the absence of oxygen,

meaning that the additional amount of organic compounds that was extracted under anoxic conditions compared to oxic conditions contain more oxidized functional groups.

### 3.3 Effects of different organic matter extracts on rate and extent of microbial ferrihydrite reduction and mineral transformation during reduction

To determine the effects of SOM, FA and HA extracts on microbial Fe(III) reduction, the Fe(III) mineral ferrihydrite was incubated with the Fe(III)-reducing bacterium *Shewanella oneidensis* MR-1 in setups amended with our extracts and total Fe(II) was monitored over time (Fig. 3). AQDS, i.e. 2,6-anthraquinone disulphonate, a quinone model compound commonly used in electron shuttling studies that can significantly increase the extent of microbial Fe(III) reduction, was used as a





reference for a significant stimulation of Fe(III) reduction by our extracted OM via electron shuttling. The highest initial

microbial Fe(III) reduction rates were determined as shown in the supporting information (Fig. S4). The presence of AQDS stimulated ferrihydrite reduction to Fe(II) with a maximum reduction rate of $3.12\pm0.38$ mmol Fe(II) $d^{-1}$ compared to setups without electron shuttle with a rate of $0.79\pm0.31$ mmol Fe(II) $d^{-1}$ (Fig. 3(a), Fig. S4). The observed decrease of total Fe(II) after 5 days of incubation (from 14.67 mM to 6.87 mM) in the AQDS-amended setup was caused by Fe(II) loss due to sorption of Fe(II) or precipitation of Fe(II) (e.g. as Fe(II)-phosphate mineral due to the presence of phosphate buffer) at the

wall of the glass bottles (Fig. S2). After the addition of oxically and anoxically water-extracted SOM (Fig. 3(d)), Fe(III) was reduced at maximum rates of $1.53\pm0.20$ mmol Fe(II) $d^{-1}$ and $2.07\pm0.43$ mmol Fe (II) $d^{-1}$, respectively, suggesting higher reduction rates than without any electron shuttle ($0.79\pm0.31$ mmol Fe(II) $d^{-1}$).

When comparing Fe(III) reduction in the presence of the different HA extracts (Fig. 3(e), 3(f)), we found that amendment with HA chemically-extracted with NaOH from the soil under anoxic conditions showed the fastest reduction rate

($1.83\pm0.03$ mmol Fe(II) $d^{-1}$) followed by HA isolated from the water extract anoxically ($1.70\pm0.25$ mmol Fe(II) $d^{-1}$) and HA chemically extracted from soil oxically ($1.55\pm0.08$ mmol Fe(II) $d^{-1}$). The reduction rate of the setup amended with HA isolated oxically from the water extract was $0.82\pm0.27$ showing slight stimulation effect compare to the setup without electron shuttle ($0.79\pm0.31$mmol Fe(II) $d^{-1}$). Addition of FA increased Fe(III) reduction rates significantly in all cases (Fig. 3(b), 3(c)). In the presence of FA isolated oxically and anoxically from water-extracted SOM, the fastest rates were

$2.03\pm0.54$ and $2.22\pm0.36$ mmol Fe(II) $d^{-1}$, respectively. After addition of FA chemically-extracted from the soil under oxic and anoxic conditions, the maximum reduction rates were even faster with $2.31\pm0.15$ and $3.05\pm0.07$ mmol Fe(II) $d^{-1}$. Control samples with only OM and ferrihydrite (without bacteria) did not show any ferrihydrite reduction (Fig. S3).

In addition to differences in reduction rates depending on the identity of the added organic extract, we also found differences in reduction extents. In most cases, the reduction extent was higher in the presence of OM compared to OM-free setups

(3.87 mM of Fe(II) after 15 days). Specifically, setups amended with FA showed higher microbial Fe(III) reduction extents than with HA. After 25 days of incubation, setups with FA extracted chemically from soil under anoxic conditions reduced 10.87 mmol $L^{-1}$ Fe(III) to Fe(II), while the maximum Fe(III) reduction extent in the presence of added HA (chemically isolated from anoxically water-extracted SOM) was about 7.08 mmol $L^{-1}$ Fe(II) (Fig. 3(c) and (e)).

Since the used OM extracts contained some Fe(II) and Fe(III), we evaluated the contribution of these ions to the observed

Fe(III) reduction (Tab. S3). First, the Fe(II) present in the water-extracted SOM, FA and HA ranged from 7.2 (water-extracted FA, oxic) to 79.2 $\mu$mol $L^{-1}$ (SOM, water-extracted, anoxic) (Tab. S3) and made up between 1-17.6% of the measured Fe(II) concentration after 30 minutes of incubation. With the increase of Fe(II) concentration over time, the percentage of Fe(II) present in the extracts to the measured Fe(II) concentration decreased to less than 0.1% and is therefore negligible. Second, the influence of Fe(III) initially present in the water-extracted SOM, FA and HA (Tab. S2) can be



neglected as well, because the Fe(III) concentration of the extracted organic matter fractions ranged from 8.7-43.9 $\mu$mol L$^{-1}$ (Tab. S3), but the ferrihydrite concentration used in the setups was 15 mmol L$^{-1}$.

## 4 Discussion

### 4.1 Effects of the presence of oxygen on the amount and properties of SOM extracts

The presence and absence of oxygen impacted the amount of water-extractable SOM. Under anoxic conditions, water at neutral pH extracted about 6.7 times more organic carbon than under oxic conditions (Tab. 1). The presence of Fe(II) at the end of extraction in all anoxic extracts suggested that the higher amount of extracted OM is probably related to microbial Fe(III) mineral reduction and the release of mineral-bound OM during mineral dissolution. A correlation between the dissolved organic carbon (DOC) concentration and the amount of Fe(II) in pore water was reported before for sediment

samples that were incubated in the dark under anoxic conditions for 14 days (Dadi et al., 2017). Other studies also suggested an increase in DOC under anoxic conditions due to the microbial iron(III) mineral reduction and dissolution and the concomitant release of organic carbon (OC) that was co-precipitated with and adsorbed to the iron(III) minerals(Gu et al., 1994; Riedel et al., 2013; Shimizu et al., 2013).

In addition to differences in the amount of extracted OM, the presence or absence of oxygen also influenced the aromaticity

of the extracted SOM, as shown by the SUVA$_{254}$ values (Tab. 1), the EEMs, (Fig. S1) and to some extent also by the $^{13}$C-NMR data (Tab. S2). The SOM extracted anoxically showed a higher aromaticity, suggesting that the additional organic matter mobilized by reductive dissolution of iron minerals possesses a higher degree of aromaticity. This is in line with findings described by other studies (Gu et al., 1994; Lv et al., 2016; Avneri-Katz et al., 2017; Coward et al., 2019). Kothawala and co-authors (Kothawala et al., 2012) incubated oxically-extracted soil solution with soils with different

mineral composition. SUVA and fluorescence index analysis of the remaining non-sorbed organic matter showed that regardless of the soil type, the aromatic functional groups were preferentially adsorbed to the soil minerals.

### 4.2 Effect of extraction pH on the amount and properties of extracted organic matter

The practice of extracting and isolating HA and FA using NaOH and HCl under anoxic conditions has been the established

standard protocol (IHSS, 2017). As early as in 1972, Swift and Posner (Swift and Posner, 1972) showed that by incubating a peat HA with 1 M NaOH under oxic conditions for 30 days, more than half of the HA was degraded to low-molecular-weight molecules and amino acid N was lost from the HA. Later studies also reported the hydrolysis of esters in NOM to carboxylic acid groups when exposing NOM to NaOH under oxic conditions (Ritchie and Perdue, 2008). Consistent with previous studies, our SUVA$_{254}$ (Tab. 1), EEM (Fig. S1), $^{13}$C-NMR (Tab. S2) and EEC (Fig. 2) results showed lower



aromaticity and EEC of chemically-extracted FA and HA under oxic conditions compared to chemically-extracted FA and HA extracted under anoxic conditions, indicating the degradation of aromatic structures and functional groups in the OM to smaller molecules in the presence of $O_2$ under high pH conditions.

However, we found that even under anoxic conditions, the chemical extraction extracted up to 100 times more carbon than the water extraction at neutral pH (Tab. 1), consistent with previous studies (Aiken, 1985). This higher extraction efficiency

at high pH could be due to the deprotonation of carboxyl and phenol functional groups leading to both higher aqueous solubility and electrostatic repulsion of OM from negatively charged soil minerals (Kleber et al., 2015) or due to the hydrolysis of plant material and the formation of smaller oligo- and monomers (Sparks, 2003). Not only the amount of C extracted but also the properties of the extracted FA and HA are affected by the chemical extraction under anoxic conditions. Our results indicate that the chemically-extracted HA have higher aromaticity than water-extracted SOM under anoxic

extraction conditions. On the one hand, the higher aromaticity in chemically-extracted HA can probably be explained by the extra amount of C extracted from soil by the chemical extraction method. On the other hand, this cannot be the only explanation, since the HA chemically separated from the water-extracted SOM (water-extracted HA) also had higher aromaticity than the water-extracted SOM itself. This suggests the formation of aromatic functional groups during the extraction with NaOH under anoxic conditions by condensation reactions between amino acids, aldehydes, hydroxyl-and

catechol-containing molecules. Such condensation reactions could result in larger molecules with a higher degree of aromaticity (Gieseking, 1975; Golchin et al., 1994; Kappler and Brune, 1999; Kappler and Haderlein, 2003).

A recent study comparing OM extraction from a freshwater sediment using water (acidified to pH 2 with 1 M HCl), with an extraction using 0.1 M sodium pyrophosphate (pH 10) and 0.5 M NaOH (pH 12) also revealed a higher aromaticity in the alkali-extracted OM (Fox et al., 2017). Using Fourier-transform infrared spectroscopy (FTIR) and electrospray ionization

Fourier-transform ion cyclotron resonance spectrometry (ESI-FTICR-MS), these authors showed that OM extracted by sodium pyrophosphate and NaOH had more condensed aromatic compounds. However, whether the chemically-extracted HA has higher aromaticity than water-extracted SOM due to the usage of NaOH has been questioned as well (Golchin et al., 1994; Olk et al., 2002). Recently, Olk and co-authors collected water samples from Suwannee River (Georgia, USA), extracted HA following the IHSS protocol and natural organic matter (NOM) by using the reverse osmosis (RO) method that

includes no usage of chemicals. By using electrospray ionization Fourier-transform ion cyclotron resonance mass spectrometry (FT-ICR-MS), they found that the molecular formulas of the HA and SOM extracts had a similarity of 66%. Therefore, they believed that very minor chemical modifications of NOM happened during the chemical isolation procedure (Olk et al., 2019). In contrast, in another study (Li et al., 2016), NOM was also extracted from Suwannee river using a styrene-divinylbenzene copolymer (PPL)-based solid phase extraction (SPE) (PPL-SPE_NOM) and the same RO method as

used in Olk et al. (2019) (RO-NOM). [1]H-NMR analysis was conducted to compare the chemical structure of these two NOM extracts to the original Suwannee river samples, and it was found that the RO-NOM extracts had a remarkably higher





abundance of aromatics and carboxyl-rich alicyclic molecules and less aliphatic groups comparing to the Suwannee river samples or the PPL-SPE-NOM extracts. These findings rise the question whether the RO extraction is selective to aromatic and alicyclic carbons. Finally, it is well known that aquatic NOM (e.g. from Suwannee river) differs a lot form soil NOM

regarding the average molecular size and chemical structure (i.e., the aromaticity) (Chen et al., 2003a). Therefore, even if the HA chemically extracted from the Suwannee river had similar molecular properties than the Suwannee-river NOM itself, this still does not mean that in soil environments there is no significant differences in aromaticity between the differently extracted OM fractions.

**4.3 Electron exchange capacity (EEC) of soil extracts determines their ability to stimulate microbial Fe(III) reduction**

Our data showed that the rates of microbial ferrihydrite reduction differed in the presence of different OM extracts. The observed differences in Fe(III) reduction rates can either be a result of the differences in OM redox activity (e.g. number and redox potentials of redox-active functional groups) and the resulting function of the OM as electron shuttle or due to different secondary mineral phases that can form during ferrihydrite reduction. However, a previous microbial Fe(III)

mineral reduction study of 5 mM ferrihydrite in the presence of 50 mg C $L^{-1}$ OM, 0.8 mM phosphate buffer and $2\times10^5$ cells mL$^{-1}$ *Shewanella oneidensis* MR-1 showed no goethite or magnetite (based on $^{57}$Fe-Moessbauer and XRD analysis) but vivianite as the major mineral phase (Amstaetter et al., 2012). The transformation of ferrihydrite to vivianite instead of goethite or magnetite in presence of phosphate buffer was also reported in other studies using similar concentration of OM, buffer, cells and ferrihydrite (Chen et al., 2003a; Piepenbrock et al., 2011; Shimizu et al., 2013). The formation of more

crystalline secondary mineral phases such as goethite was only observed during ferrihydrite reduction in the absence of phosphate (Hansel et al., 2003; Borch et al., 2007). Abiotic experiments showed that phosphate inhibits the transformation of ferrihydrite to magnetite or goethite by blocking of surface sites of ferrihydrite, therefore prevents the sorption of the produced Fe(II) on the Fe(III) mineral, thus lowering the number of surface sites where conversion of ferrihydrite to magnetite or goethite can take place (Galvez et al., 1999). Therefore, the transformation of ferrihydrite to magnetite or

goethite is not expected to happen in our experiments and the following discussion will focus on the influence of the redox activity of the extracted OM on the rate and extent of the microbial Fe(III) reduction.

As measures for the redox activity of the different extracted OM fractions, we determined their potential for accepting electrons (EAC) and for accepting and donating electrons (EEC). Correlating the EAC values of our different OM extracts and the maximum microbial ferrihydrite reduction rates showed that the higher the EAC values of the extracted SOM, FA

and HA, the faster the microbial Fe(III) reduction rates are (Fig. 4). As shown before (Aeschbacher et al., 2012), (hydro)quinone functional groups contribute mainly to the measured EAC values in OM and these quinone moieties are thought to be the major functional group responsible for electron transfer between Fe(III)-reducing bacteria and Fe(III) minerals during electron shuttling. Scott et al. (1998) reported a direct correlation between OM oxidation capacity and the





stable free-radical content in the OM, stemming from semiquinone radicals (Lovley et al., 1996; Scott et al., 1998). However,

we also found a correlation between EEC values and the maximum microbial ferrihydrite reduction rates in the presence of

SOM, HA and FA. Since higher EEC values reflects higher contents of aromatic/polycondensed aromatic compounds in the

OM (Aeschbacher et al., 2012), our results also indicate that, apart from quinones, also other aromatic functional groups

were involved in the microbial Fe(III) reduction with OM as electron shuttles and these functional groups also influence the

electron transfer efficiency between the Fe(III)-reducing bacteria, the OM and the Fe(III) minerals. Support for the

participation of non-quinone groups in such OM electron transfer studies also comes from previous analyses of redox

properties and stable free-radical concentrations in OM (Struyk and Sposito, 2001; Chen et al., 2003a).

Faster Fe(III) mineral reduction rates in the presence of more aromatic functional groups (including quinones) was

demonstrated previously in experiments with increasing concentrations of AQDS or HA (Jiang and Kappler, 2008; Wolf et

al., 2009; Glasser et al., 2017). It was suggested that the microbial turnover of substrate (lactate as electron source) is limited

by the availability of the electron acceptor, i.e. either by the Fe(III) in the absence of shuttles or by the OM when OM serves

as electron shuttling compound (Jiang and Kappler, 2008; Poggenburg et al., 2018). Thus, with the same concentration of

OM electron shuttle, the OM with more redox-active functional groups can accept more electrons per time from the

microorganisms, therefore resulting in higher Fe(III) reduction rates. Additionally, when more quinone or other redox-active

functional groups are present per shuttle molecule, the distance between redox-active functional groups is smaller, therefore,

electron transfer within the shuttle molecule and between the shuttle molecules can occur faster, thus further increasing the

electron transfer rate from the microbial cells to the shuttle molecules and further to the Fe(III) minerals (Boyd et al.,

2015;Glasser et al., 2017). The different types and proportions of functional groups in the different OM extracts may also

influence their adsorption onto the ferrihydrite surface, and therefore also impact the rates of microbial ferrihydrite reduction

amended with different OM. However, due to the high concentration of lactate and HEPES buffer in our experiment, we

could not quantify the amount of adsorbed OM vs. dissolved OM. It has to be noted, however, that in our extracted OM (Fig.

2) different amounts of redox-active Fe ions were present and that the redox-active OM-bound Fe can potentially also

influence the rates of Fe(III) mineral reduction. The OM-bound Fe(III) can also be reduced by the Fe(III)-reducing bacteria

or by reduced organic functional groups in the OM to Fe(II), which can then transfer electrons further to the ferrihydrite. The

OM-bound Fe is subsequently reoxidized to Fe(III), and therefore contributes to electron shuttling between Fe(III)-reducing

bacteria and ferrihydrite.

In addition to the differences in the reduction rates, we also observed that the extent of ferrihydrite reduction was influenced

by the presence of different OM. Specifically, amendments with HA lead to lower extents of Fe(III) reduction than FA

amendments. This difference could be caused by the higher content of aromatic functional groups in HA than in FA and the

resulting differences in sorption properties. OM with higher aromaticity and larger molecular weight was shown to have a

higher absorption affinity to ferrihydrite (Lv et al., 2016; Coward et al., 2019). Since our HA extracted under all conditions





were more aromatic than the FA, the HA were probably preferentially adsorbed to ferrihydrite. On the one hand, the sorbed HA can block surface sites on the minerals and restrict the accessibility for bacteria. On the other hand, HA adsorption changes the net surface charge of ferrihydrite from positive to (partially) negative and thus leads to repulsion of negatively charged cells (Aeschbacher et al., 2012).


### 5 Conclusions

In summary, our results clearly show that the extraction method determines the concentration of redox-active (aromatic) functional groups and the EEC of the soil extracts and the EEC is a key factor for the electron shuttling capacity of soil extracts in microbial Fe(III) mineral reduction. Therefore, it has to be carefully decided which SOM extraction method to

apply and which SOM fraction to use in biogeochemical experiments to obtain soil extracts that can represent natural SOM. Representative SOM is necessary to obtain meaningful results that will prevent overestimating the reactivity of SOM in redox processes in the environment. Based on our experimental results we suggest that firstly, the NaOH extraction method should be avoided in general because it alters the chemical and redox properties of SOM. Additionally, soil pH values typically range from 3.5-8.5 (Sparks, 2003), therefore the organic matter that is soluble only at pH>12 will not be dissolved

under in-situ soil conditions and might react differently in biogeochemical processes compared to solid-phase soil OM (Roden et al., 2010; Kappler et al., 2014). Secondly, when extracting SOM with water at neutral pH, the redox milieu (oxic or anoxic) during extraction needs to be carefully controlled. When targeting oxic environmental systems with the goal of obtaining relevant OM matter that participates in biogeochemical processes under such redox conditions, short extraction times (<24 h), small batches, aeration, and thorough stirring is recommended for the OM extraction. Thus, anoxic conditions

during the OM extraction should be avoided that would lead to reductive dissolution of iron minerals with concomitant mobilization of OM that would not be available under oxic conditions (in the absence of microbial Fe(III) reduction). However, in case the target environmental systems undergo redox fluctuations or even permanent reducing conditions, yielding anoxic conditions with microbial Fe(III) mineral reduction during the OM extraction is appropriate.

*Data availability*. Raw data of all results presented in this study are available and can be found on https://issues.pangaea.de/browse/PDI-21233.

*Supplement*. Supporting information includes the statistical analysis of the specific UV absorbance at 254 nm (SUVA$_{254}$) data, the [13]C-NMR analysis, the Fe(II) and Fe(III) content and their contribution to the electron exchange capacity (EEC) of the OM extracts, the statistical analysis of the EEC data, the fluorescence excitation-emission (EEM), the picture showing the sorption and precipitation of

Fe(II) on the wall of glass bottles of the microbial Fe(III) reduction experiments containing AQDS, the results of the abiotic reduction of



Fe(III) minerals with the addition of OM extracts, and the calculation of the rates of microbial Fe(III) reduction amended with different OM extracts. The supporting information is available online at XXX.

*Author contribution*. YB, AK, SH designed the experiment, YB conducted the experiments and analyzed the data of OM extraction, fluorescence spectra and microbial Fe(III) reduction together with AK. ES performed and analyzed data of the electron chemical analysis

and helped with the fluorescence spectra. HK conducted the NMR analysis and processed the data. YB and AK prepared the paper, with great help from SH.

*Acknowledgement*. We acknowledge Prof. Scholten (Univ. Tuebingen) for his help with soil sampling. We thank Ellen Röhm, Lars Grimm, Franziska Schaedler for technical support, Alison Enright, Zhe Zhou and Zhen Yang for their help on improving the manuscript.

*Financial support*. This study was supported by a grant from the German Research Foundation (DFG) to AK (KA1736/37-1)




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





**Table 1.** Total dissolved organic carbon (DOC) concentration of the water extracted SOM and chemically extracted FA, total amount of carbon in all extracted SOM, HA and FA fractions, as well as the calculated percentage of carbon extracted from the soil. DOC of SOM and FA fractions was quantified directly from their extracted solutions, whereas the FA chemically isolated from water-extracted SOM and the precipitated HA were first freeze-dried and then re-dissolved in Milli-Q water at pH 7 for analysis. The difference between total carbon (mg) of water-extracted

5 SOM and the sum of total carbon (mg) of water-extracted FA and water-extracted HA (under oxic and anoxic conditions) is due to the loss of material during the isolation of FA and HA. Values are means±standard deviation (SD) of triplicates. An unpaired two-sided T-test was done to analyze the statistical significance of each pair of samples that was treated with the same extraction method but under different conditions (i.e., water-extracted FA, oxic, was compared to water-extracted FA, anoxic)

| | SOM | | FA | | | | [a]HA | | | |
|---|---|---|---|---|---|---|---|---|---|---|
| | Water-extracted, oxic | Water-extracted, anoxic** | [a]Water-extracted, oxic | [a]Water-extracted, anoxic** | Chem.-extracted, oxic | Chem.-extracted, anoxic** | Water-extracted, oxic | Water-extracted, anoxic** | Chem.-extracted, oxic | Chem.-extracted, anoxic** |
| DOC concentration in extract (mg C L$^{-1}$) | 0.149± 0.036 | 0.890± 0.041 | – | – | 5.800± 0.025 | 6.320± 0.071 | – | – | – | – |
| Total organic carbon in extract (g)[b] | 0.036± 0.012 | 0.234± 0.015 | 0.021± 0.002 | 0.146± 0.013 | 1.451± 0.008 | 1.770± 0.028 | 0.014± 0.003 | 0.079± 0.000 | 1.450± 0.002 | 1.881± 0.029 |
| Percentage of carbon extracted from soil (%)[c] | 0.41±0.14 | 2.74±0.18 | 0.24± 0.02 | 1.69± 0.15 | 17.0± 0.09 | 20.7±0.32 | 0.15± 0.03 | 0.90± 0.00 | 17.0±0.02 | 22.0±0.34 |
| [d]SUVA$_{254}$ (mg$^{-1}$ C cm$^{-1}$) | 0.018 | 0.027 | 0.017 | 0.029 | 0.023 | 0.042 | 0.068 | 0.207 | 0.083 | 0.265 |

[a] TOC of all HA extracts and water-extracted FA were directly measured from the freeze-dried powders

10 [b] Total carbon (g) = TOC (g C L$^{-1}$) × volume of the extracted solution (L)

[c] Percentage of carbon extracted from soil = Total carbon extracted (mg g$^{-1}$)/soil carbon content (8.54 mg g$^{-1}$)

**The percentage of carbon extracted from soil of the samples are significantly different (n=2, two-sided t-test, $P < 0.05$

[d] Specific UV absorbance 254 nm (SUVA$_{254}$) = UV254 ×DOC (mg C L$^{-1}$), b is the optical path length in centimeter (1 cm in this experiment)



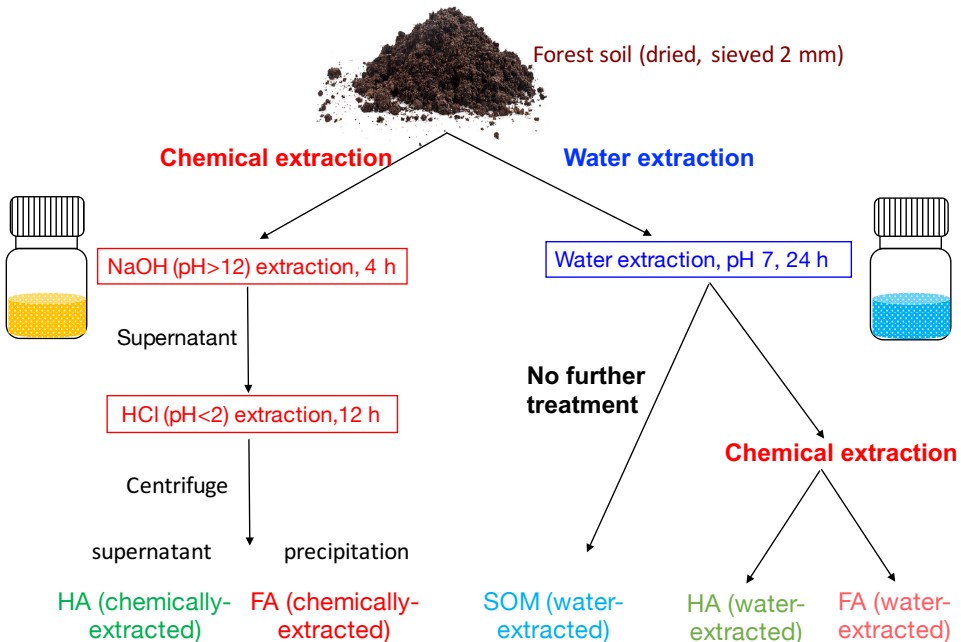

**Figure 1.** Schematic diagram of soil organic matter (SOM) extraction process. Forest soil samples (Schönbuch forest, Baden-Wuerttemberg, Germany) was dried under 30 ˚C and grinned to pass 2 mm sieve. To chemically extraction FA and

5    HA, 100 g soil was incubated with 400 mL 0.1 M NaOH (pH 12) for 4 h, after centrifugation, the supernatant was acidified by HCl to pH<2, HA then precipitated out thus be separated from FA. For water extraction, 100 g soil was incubated with 400 mL doubly-deionized water (<18.2 MΩ.cm; Milli-Q, Millipore) at pH 7. pH was monitored during the extraction and it remained stable (range between 7.02-7.33). The slurry was centrifuged after 24 h; we define the supernatant as water extract. A 200 mL aliquot of the supernatant containing the water-extracted SOM was amended with 1 M NaOH until pH 12. After

10    4 h of incubation, the pH of the solution was adjusted to <2 by 6 M HCl to precipitate HA and to separate FA. This extraction was conducted under both oxic and anoxic conditions.





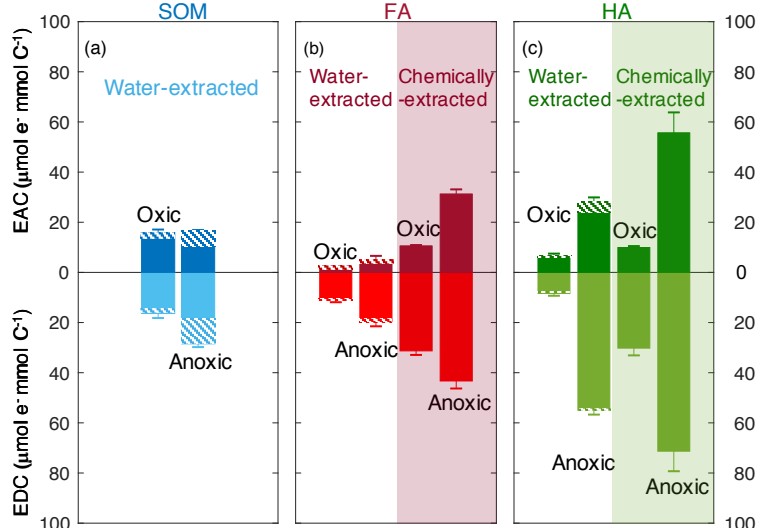

| | EEC (mmol e⁻ mmol C⁻¹) | *Carbon content (mmol) | Total number of electrons that can be exchanged (mmol e⁻) |
|---|---|---|---|
| FA (water-extracted, oxic) | 13 | 3 | 39 |
| HA (water-extracted, oxic) | 15 | 19 | 285 |
| SUM | | | 324 |
| SOM (water-extracted, oxic) | 32 | 2 | 64 |
| | | | |
| FA (water-extracted, anoxic) | 24 | 12 | 288 |
| HA (water-extracted, anoxic) | 83 | 6 | 498 |
| SUM | | | 786 |
| SOM (water-extracted, anoxic) | 44 | 19 | 836 |

*carbon content was calculated from the total organic carbon content in extracts in Tab. 1

**Figure 2.** Electron exchange capacity (EEC), the sum of EAC (electron accepting capacity) and EDC (electron donating capacity) of SOM (a), FA (b) and HA (c) extracted from Schönbuch forest soil under oxic or anoxic conditions. Areas with white background represent water-extracts (without further treatment in (a) and after NaOH and HCl treatment in (b) and (c), respectively) whereas red and green shaded areas represent chemically-extracted FA and HA, respectively. Hashed areas represent the EAC or EDC contribution stemming from redox-active Fe(III) and Fe(II) ions in the samples. The integrated current response was normalized to the DOC of all extracts. Despite the contribution of Fe to the redox activity, we decided to normalize the EAC/EDC values to C content first because in most cases the Fe contribution was small and second because





normalizing the EAC/EDC values to total weight of material would be misleading since the fraction of inorganic, non-redox active constituents (e.g. $Mg^{2+}$, $Ca^{2+}$ ions from salts present in the soil) varies significantly between the different extracts (Tab. S5). Error bars indicate the standard deviations of at least 4 replicates. 2-way ANOVA statistical analysis was conducted and the result suggested the measured EEC values of the extracts were significantly different from each other

5 (P<0.005, Tab. S4). The table underneath the figure shows the recovery of total number of electrons that can be exchanged (that means transferred from Fe(III)-reducing bacteria to the OM, or from the OM to Fe(III) minerals) by water-extracted SOM before and after the chemical separation of the SOM extract into HA and FA.


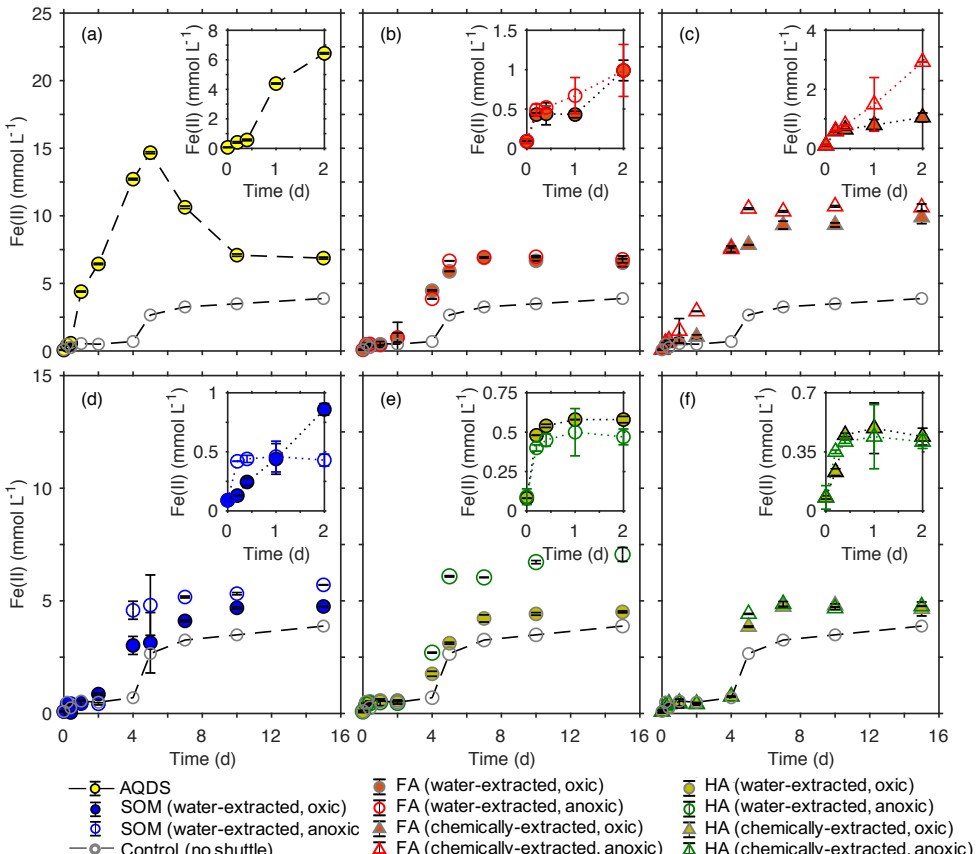

**Figure 3.** Microbial reduction of ferrihydrite (15 mmol L$^{-1}$) by *S. oneidensis* MR-1 (10$^7$ cells mL$^{-1}$) in the presence of 15 mmol L$^{-1}$ lactate as electron donor and 50 mg C L$^{-1}$ FA (b, c), SOM (d), and HA (e, f) compared to 100 μmol L$^{-1}$ AQDS (a) presented as formation of total Fe(II) over time. The inserts in panels a-f show the data points for the first 6 h of incubation. All setups were incubated in air-tight 100 mL glass serum bottles flushed with N$_2$ at 30°C in the dark. Control samples were incubated at the same condition in the absence of electron shuttles (ferrihydrite, lactate, and cells only). Error bars represent standard deviations of triplicate bottles.

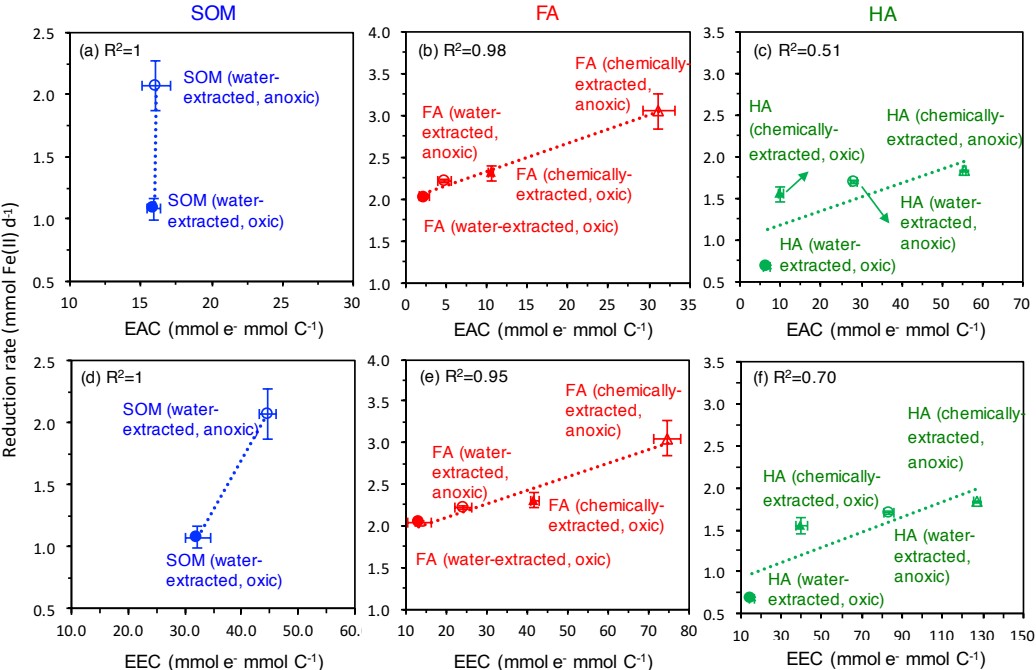

5 **Figure 4.** Correlation of the electron exchange capacity (EEC) (bottom) and electron accepting capacity (EAC) (top) with the fastest microbial ferrihydrite reduction rates (*Shewanella oneidensis* MR-1) in the presence of oxically and anoxically prepared SOM (a), FA (b) and HA (c). EEC and EAC values are re-plotted from Fig. 2. Please note that the EEC and EAC values were determined by the electrochemical method described in the methods section and represent the contribution of both redox-active organic functional groups such as quinones and the redox-active Fe ions in the SOM, FA and HA extracts.

10 Horizontal error bars represent standard deviations of the measured EEC values, vertical error bars are standard deviations of the reduction rates calculated as shown in Fig. S4.