# Peer review of "High pH and anoxic conditions during soil organic matter extraction increases its electron exchange capacity and ability to stimulate microbial Fe(III) reduction by electron shuttling"

_Biogeosciences, 2019_

## Referee Comment (RC1) · Anonymous Referee #1 · 2 Dec 2019

Bai et al. compare redox properties of a water extract with humic and fulvic acid extracts of a soil. Additionally, fulvic acids and humic acids were extracted of the water extract. All extractions were performed under air as well as under N2.

The final conclusion is that (oxic or anoxic) alkaline extracts are not suitable to substitute natural organic matter in redox-experiments.

Although the concept of humic and fulvic extractions was often challenged, the method is still popular in current papers and textbooks, sometimes even described as the

method of choice. The topic is therefore both important and in time. Also, I appreciate such a clear statement from groups which frequently used humic acids in the past. The paper is well written and nicely illustrated by figures and tables.

One general comment: In the manuscript the abbreviation "SOM" is used to address the water extractable fraction (tables, figures, text). This seems misleading to me: SOM comprises all non-living soil organic matter, while a water extract is done to gain something similar to the dissolved organic matter or the fraction which can be dissolved or mobilized (colloidal fraction) during rain events or rising groundwater. I think, a better name for the water extract would be "water extract (WE)" or "water extractable organic matter (WEOM)" instead of SOM. WE or WEOM also shows that one is aware of the difference between a water extract and the real DOM of a soil solution.

To my knowledge, humic substances are meant to extract all HA and FA of a sample, while it is accepted that a large fraction may remain behind. I therefore wonder if it is meaningful to compare extracted C amounts from alkaline extraction with water extraction. With respect to the different extraction of functional groups or artefacts during exposure to pH12, the comparison between WEOM and FA/HA of WEOM are certainly more robust. (For example, does the sum of EAC(FA of WEOM, oxic) and EAC (HA of WEOM oxic) equals EAC(WEOM) or do we see artefacts? The sum of FA of WEOM+HA of WEOM (0.035 g C) is very close to the total WEOM (0.036 g C)).

Other comments:

First sentence: I propose to delete "including humic substances". The main point can be made without and if alkaline extraction causes that many artefacts, it may even be wrong.

Line 15: name soil, sampled depth/horizon and pH of the soil.

Line 18/19: ". . .100 times more. . ." see general comment above.

Line 25: . . .changes in functional groups . . . is there space to name observed or as-

sumed changes in the abstract?

Line 27: Delete "at neutral pH". Rainwater has a pH of 5, soil solutions are rather variable in pH. The choice of pH should therefore be adopted to the soil and the specific research question. Ionic strength and ionic composition will also play a role. Was pH7 a good choice for the Cambisol?

Line 81: Was the mineral topsoil horizon sampled or the mineral topsoil plus litter layer? Do the upper 15 cm include material of the B horizon? Is anything known about the Fe oxide content of the soil? If yes, does the amount of Fe oxides fit to the observed Fe concentration in the extracts?

Line 63: please give ferrihydrite concentration in mM Fh-Fe or in weight %. As the composition of ferrihydrite is still under debate and as a mineral structure can be given in different ways, mM Fh is not unambiguous.

Line 194: delete the "e"

Line 211: please correct, the table in Fig.2 give 44 $\mu$mol e- mmolC-1, instead of 45.

Line 304-307: sentence?

Line 400: "adsorption" instead of "absorption"

Table 1 is unclear. What was analyzed, DOC (after 0.45 filtration) or TOC? Why are concentrations and masses of C needed? Can the descriptive first and second rows of the table be optimized? "Water extracted" above columns 4,5,8, and 9 is misleading, as the treatment is a chemical extraction of a water extract. The latter also applies for the figures. Replace SOM by WE or WEOM or the like.

Figure 3 is very confusing due to the complex legend.

Figure 1, although nice, could be sacrificed to show NMR or EEMS data. It's a pity, that these ended up in the SI. Can NMR spectra be given at least in the SI?

Title: While the manuscript is concise and clear, the title is less straight. How about having the main message in the title?

---

## Author Comment (AC1) · 9 Dec 2019

see pdf file attached

Please also note the supplement to this comment:
https://www.biogeosciences-discuss.net/bg-2019-308/bg-2019-308-AC1-supplement.pdf

---

## Referee Comment (RC2) · Anonymous Referee #2 · 12 Dec 2019

This paper compares how different extraction methods influence the composition of soil organic matter (SOM) derived from the process. They compared SOM extracted by neutral pH water and mediated by alkaline extraction followed by acid precipitation (the standard approach used to delineate soil humic and fulvic acids) under oxic and anoxic conditions. The authors determined carbon recovered, specific UV absorbance @ 254 nm (SUVA), and most importantly the exchangeable electron capacity (EEC) as well as electron accepting and donating capacities (EAC and EDC). The manuscript is well-organized, and easy to read (even though there are a couple of typos that spell

checker did not catch). The most important contribution, however, is the electrochemical analyses that were conducted, which makes this paper really unique. There are a few major issues that I have, and specific comments are below.

1. The SUVA data seems fine for the water extracted SOM and fall well within the range of values reported by others (e.g., Weishaar et al., 2003). However, the FA and HA alkaline extraction conducted anoxically were off the charts and many factors higher than the highest value reported by Weishaar et al., 2003. These numbers appear unrealistic and could be due to the presence of iron (both (II) and (III)) in the extracts that reached 3 mM. Given that Weishaar et al., reported iron interference (they use Fe(III) as an example, but noted that Fe(II) can also interfere) at levels of only a few mg/L (or 10's of $\mu$M) this could be a positive interference to their SUVA data. 2. There was only passing mention of the NMR and fluorescence data. Why wasn't this data more prominently discussed in the paper (as opposed to a glancing mention in the SI)? For example, how does the EEM data "confirm higher contents of aromatic carbon" (the explanation in the SI caption was inadequate)? Further, the relatively smaller differences in NMR determined aromaticity between anoxically extracted vs oxic extraction SOM is not reflected in the much larger (order of magnitude) spread observed for SUVA (see above). Further, the EEMs from Figure S2 look really odd and I suspect that this caused by the really high DOC levels used by the authors (100 mg/L!). At those levels inner-filter-effects will become dominant as the solution will be optically dense to the point where inner-filter corrections will likely no longer work. Typically, fluorescence EEMs are collected at much lower (nearly two orders of magnitude) DOC concentrations to minimize inner-filter-effects (see papers by Stedmon et al., in L and O). Thus, because the data is likely improperly collected I would simply eliminate it from the discussion. 3. I think the discussion regarding the comparison between Suwannee River reverse osmosis dissolved organic matter (DOM) to the fulvic acid fraction isolated by XAD-8 chromatography (as opposed to acid precipitation) does not add any value to the paper because you are basically comparing apples and oranges (i.e., SOM vs. aquatic DOM). The methods are totally different from alkaline and neutral extraction and there

are no mineral phases involved. The authors can delete the entire discussion and it will not affect the conclusions or the quality of this paper. 4. While the authors point to several studies demonstrating correlations between DOC and Fe(II) formed from the dissolution of iron oxides in batch incubation studies, evidence for this relationship has also been reported in benthic pore waters. See papers by Burdige (et al.,), Chin (et al.,), plus many others. I think showing that this phenomenon occurs in real aquatic systems strengthens the arguments put forth by the authors for this paper.

---

## Author Comment (AC2) · 12 Dec 2019

This paper compares how different extraction methods influence the composition of soil organic matter (SOM) derived from the process. They compared SOM extracted by neutral pH water and mediated by alkaline extraction followed by acid precipitation (the standard approach used to delineate soil humic and fulvic acids) under oxic and anoxic conditions. The authors determined carbon recovered, specific UV absorbance @ 254 nm (SUVA), and most importantly the exchangeable electron capacity (EEC) as well as electron accepting and donating capacities (EAC and EDC). The manuscript is well-organized, and easy to read (even though there are a couple of typos that spell checker did not catch). The most important contribution, however, is the electrochemical analyses that were conducted, which makes this paper really unique. There are a few major issues that I have, and specific comments are below.

We would like to thank the reviewer for going through our manuscript and providing useful suggestions to help us to improve the manuscript. We also appreciate the overall positive response and in particular his/her appreciation of the electrochemical analyses of the SOM extracts. We will go through the manuscript carefully and incorporate all suggestions and comments.

1. The SUVA data seems fine for the water extracted SOM and fall well within the range of values reported by others (e.g., Weishaar et al., 2003). However, the FA and HA alkaline extraction conducted anoxically were off the charts and many factors higher than the highest value reported by Weishaar et al., 2003. These numbers appear unrealistic and could be due to the presence of iron (both (II) and (III)) in the extracts that reached 3 mM. Given that Weishaar et al., reported iron interference (they use Fe(III) as an example, but

noted that Fe(II) can also interfere) at levels of only a few mg/L (or 10's of µM) this could be a positive interference to their SUVA data.

We agree with the reviewer that the $SUVA_{254}$ values measured in our study, especially for the HA isolated from the water-extracted OM (0.207 L $mg^{-1}$ C $cm^{-1}$), are almost one order of magnitude higher than the reported $SUVA_{254}$ values for HA chemically extracted from Coal Creek soil in Weishaar et al., 2003 (0.039 $mg^{-1}$ C $cm^{-1}$) (Weishaar et al., 2003), and also higher than the typical $SUVA_{254}$ values of HA analyzed in many other studies (Beckett et al., 1987; Chen et al., 2003; Fox et al., 2017). However, the $SUVA_{254}$ of all our FA extracts range from 0.017 to 0.042 L $mg^{-1}$ C $cm^{-1}$, and these values are in line with previous studies (Beckett et al., 1987; Chen et al., 2003; Fox et al., 2017).

One reason for the higher $SUVA_{254}$ values for the HA isolated from water-extracted SOM under anoxic conditions in our study compared to others, in addition to the differences in the soils from which the HA were extracted, as suggested by the reviewer, could be the presence of Fe(II) and Fe(III). As shown by Weishaar et al. (2003), the presence of 4 mg $L^{-1}$ Fe(III) showed an absorbance value about 0.65 $cm^{-1}$ at 254 nm wavelength, and the absorbance increased with increasing Fe(III) concentrations. Based on this study, we can hypothesize that also in our case the presence of Fe(III) influenced the measured $SUVA_{254}$ value of the HA isolated from the water-extracted SOM under anoxic conditions. The Fe(III) concentration in the HA isolated from water-extracted SOM under anoxic condition was 33 µmol $L^{-1}$. Please note that we did not use the concentration of 3 mmol $L^{-1}$ as suggested by the reviewer, because 3 mmol $L^{-1}$ was the Fe(II) concentration determined right after the anoxic water extraction of SOM. However, all of the samples were passed through 0.45 mm syringe filters under oxic conditions

before the SUVA analyses. Therefore, a large amount of the Fe(II) was oxidized to Fe(III) and removed as particulate Fe(III) by the filtration, as explained in the manuscript **line 196-199, page 7**. The remaining Fe(II) and Fe(III) in the samples were analyzed and shown in Table S3 and the Fe(III) concentration for HA isolated from the water-extracted SOM under anoxic condition was 33 μmol L$^{-1}$.

$$\text{ μmol L}^{-1} \times 56 \text{ g mol}^{-1} = 1.848 \text{ mg L}^{-1}$$

According to Weishaar et al. (2003), 1.848 mg L$^{-1}$ Fe(III) has an absorbance value of 0.15 cm$^{-1}$ at 254 nm wavelength. With the additional 28 μmol L$^{-1}$ Fe(II), the contribution of iron to the measured SUVA$_{254}$ value should be even higher. Therefore, we would like to add one sentence in the results section 3.1, **line 203, page 7**, as "*A previous study showed that 4 mg L$^{-1}$ Fe(III) yielded an absorbance value of 0.65 cm$^{-1}$ at 254 nm wavelength (Weishaar et al., 2003). Therefore, we believe that the high SUVA$_{254}$ value of HA isolated from the water-extractable OM (please note, this is the term used to replace 'water-extracted SOM' as suggested by reviewer 1) compared to SUVA$_{254}$ values of HA shown in previous studies could be caused by the presence of Fe(II) and Fe(III) in the sample due to the microbial Fe(III) reduction that occurred under the anoxic extraction conditions.*"

2.  There was only passing mention of the NMR and fluorescence data. Why wasn't this data more prominently discussed in the paper (as opposed to a glancing mention in the SI)? For example, how does the EEM data "confirm higher contents of aromatic carbon" (the explanation in the SI caption was inadequate)? Further, the relatively smaller differences in NMR determined aromaticity between anoxically extracted vs oxic extraction SOM is not reflected in the much larger (order of magnitude) spread observed for SUVA (see above). Further, the EEMs from Figure S2 look really odd and I suspect that this caused by the really

high DOC levels used by the authors (100 mg/L!). At those levels inner- filter-effects will become dominant as the solution will be optically dense to the point where inner-filter corrections will likely no longer work. Typically, fluorescence EEMs are collected at much lower (nearly two orders of magnitude) DOC concentrations to minimize inner-filter-effects (see papers by Stedmon et al., in L and O). Thus, because the data is likely improperly collected I would simply eliminate it from the discussion.

We would like to thank the reviewer for the comments and suggestions. First, we would like to clarify that the concentration of all samples used for the EEM analysis was not 100 mg C L$^{-1}$ In the manuscript, line 113, page 4, it says *"Freeze-dried SOM/FA/HA powders were dissolved in Milli-Q water (pH 7) at a concentration of 100 mg C L$^{-1}$ and the solutions were agitated for 12 h at 300 rpm at room temperature, samples were then filtered through 0.45 mm syringe filter (mixed cellulose ester (MCE, Millipore, Germany). For fluorescence analyses, samples were prepared by stepwise dilution of extract solution with Milli-Q water (pH 7) until absorbance values of 0.300 at 254 nm wavelength were reached"*. Therefore, after the stepwise dilution, the final concentration of samples used for the EEM analyses was much lower than 100 mg C L$^{-1}$ and inner-filter effects can be neglected.

However, as the reviewer pointed out, we did not draw any conclusions directly from the EEM data. As also explained in the reply letter to reviewer 1, we did this because there are many debates about whether, to any extent, the EEM spectra can reflect the redox state and the aromaticity of the OM samples (Fimmen et al., 2007; Maurer et al., 2010). Since only very briefly mentioned the results of the EEM spectra and since leaving the EEMs out completely will not impact the conclusion of the manuscript at all, as

suggested by the reviewer, we would like to completely remove the EEM results from our manuscript.

Regarding the NMR, as we commented already in the reply letter to reviewer 1, we would like to show in the revised manuscript the NMR spectra of all samples in the supporting information and add the aromaticity values of the OM extracts calculated from the NMR analyses to Table 1.

Finally, regarding the reviewer question "why the very distinct difference between different OM extracts as shown by the SUVA results could not be seen in the calculated aromaticity from NMR": in previous studies that applied NMR to characterize the aromaticity of OM, the differences between different OM samples are usually in the range of 10%. For example, in the study of Lorenz et al. (2006), seven different OM samples extracted from different sampling sites were analyzed, and the aromaticity of these samples ranged from 21-32%. Inbar and co-authors compared the aromaticity of native SOM and the same SOM after 147 days of composting, and the aromaticity only changed from 35% to 37% (Inbar et al., 1990). In our study, all of the OM extracts, although extracted in different ways, were from the same soil. Therefore, we believe that, for example, the 4% difference in the aromaticity between chemically-extracted HA under oxic and chemically-extracted HA under anoxic condition does indicate a potential difference in the aromaticity of these two samples. Moreover, although the differences of the aromaticity among different OM samples calculated from NMR are not as significant as the differences of $SUVA_{254}$ values among different samples, the aromaticity of SOM/HA/FA extracted under anoxic conditions was higher than the aromaticity of SOM/HA/FA extracted under oxic conditions. Furthermore, under anoxic conditions, FA

and HA isolated from the water-extracted SOM (water-extractable OM) both have higher aromaticity than the water-extractable OM itself. Therefore, our NMR results are perfectly in line with the SUVA$_{254}$ results and the electron exchange capacity analysis of the OM samples and can be used to support and strength our argument that the chemical extraction and the presence of oxygen impacts the aromaticity thus the redox activity of the SOM extracts.

3. I think the discussion regarding the comparison between Suwannee River reverse osmosis dissolved organic matter (DOM) to the fulvic acid fraction isolated by XAD-8 chromatography (as opposed to acid precipitation) does not add any value to the paper because you are basically comparing apples and oranges (i.e., SOM vs. aquatic DOM). The methods are totally different from alkaline and neutral extraction and there are no mineral phases involved. The authors can delete the entire discussion and it will not affect the conclusions or the quality of this paper.

    Following the reviewer's suggestion, we would like to remove this part of the discussion: **line 326-343, page 12**.

4. While the authors point to several studies demonstrating correlations between DOC and Fe(II) formed from the dissolution of iron oxides in batch incubation studies, evidence for this relationship has also been reported in benthic pore waters. See papers bu Burdige (et al.,), Chin (et al.,), plus many others. I think showing that this phenomenon occurs in real aquatic systems strengthens the arguments put forth by the authors for this paper.

    As suggested by the reviewer, to strengthen our argument of the correlation between DOC and Fe(III) mineral dissolution, we would like to add one sentence as follows (at **line 285, page 10**):

*"In-situ monitoring of the DOC flux in pore water of marine sediment or freshwater wetland also suggested an increase in DOC with increasing amount of microbial iron(III) mineral reduction (Burdige et al., 1992; Burdige et al., 1999; Chin et al., 1998)".*

**References**

Beckett, R., Jue, Z. and Giddings, J.C. (1987) Determination of Molecular-Weight Distributions of Fulvic and Humic Acids Using Flow Field-Flow Fractionation. Environmental Science & Technology 21, 289-295.

Burdige, D.J., Alperin, M.J., Homstead, J. and Martens, C.S. (1992) The Role of Benthic Fluxes of Dissolved Organic-Carbon in Oceanic and Sedimentary Carbon Cycling. Geophys Res Lett 19, 1851-1854.

Burdige, D.J., Berelson, W.M., Coale, K.H., McManus, J. and Johnson, K.S. (1999) Fluxes of dissolved organic carbon from California continental margin sediments. Geochim Cosmochim Ac 63, 1507-1515.

Chen, J., LeBoef, E.J., Dai, S. and Gu, B.H. (2003) Fluorescence spectroscopic studies of natural organic matter fractions. Chemosphere 50, 639-647.

Chin, Y.P., Traina, S.J., Swank, C.R. and Backhus, D. (1998) Abundance and properties of dissolved organic matter in pore waters of a freshwater wetland. Limnol Oceanogr 43, 1287-1296.

Fimmen, R.L., Cory, R.M., Chin, Y.P., Trouts, T.D. and McKnight, D.M. (2007) Probing the oxidation-reduction properties of terrestrially and microbially derived dissolved organic matter. Geochim Cosmochim Ac 71, 3003-3015.

Fox, P.M., Nico, P.S., Tfaily, M.M., Heckman, K. and Davis, J.A. (2017) Characterization of natural organic matter in low-carbon sediments: Extraction and analytical approaches. Org Geochem 114, 12-22.

Inbar, Y., Chen, Y. and Hadar, Y. (1990) Humic Substances Formed during the Composting of Organic-Matter. Soil Sci Soc Am J 54, 1316-1323.

Lorenz, K., Preston, C.M. and Kandeler, E. (2006) Soil organic matter in urban soils: Estimation of elemental carbon by thermal oxidation and characterization of organic matter by solid-state 13C nuclear magnetic resonance (NMR) spectroscopy. Geoderma. 130, 312.

Maurer, F., Christl, I. and Kretzschmar, R. (2010) Reduction and Reoxidation of Humic Acid: Influence on Spectroscopic Properties and Proton Binding. Environmental Science & Technology 44, 5787-5792.

Weishaar, J.L., Aiken, G.R., Bergamaschi, B.A., Fram, M.S., Fujii, R. and Mopper, K. (2003) Evaluation of specific ultraviolet absorbance as an indicator of the chemical composition and reactivity of dissolved organic carbon. Environmental Science & Technology 37, 4702-4708.

---

## Author Response (AR1)

*Replies to* the interactive comments on "Effects of extraction conditions on the redox properties of soil organic matter (SOM) and its ability to stimulate microbial iron(III) mineral reduction by electron shuttling".

**Comments by the Editor (hand-written comments on the pdf file of our manuscript)**

Title: is "stimulated microbial iron (III) mineral reduction" demonstrated?

First we would like to thank the editor for spending time to go through our manuscript and helping us to improve the manuscript. We went through the manuscript thoroughly and made sure to carefully incorporate all suggestions and comments.

We believe that our results clearly demonstrated stimulating effects of soil organic matter extracts on both the rate and extent of microbial Fe(III) reduction. As shown in Figure 3 of the manuscript, after addition of only *Shewanella oneidensis* MR-1 (no soil organic matter added), after 16 days of incubation, about 3.87 mmol $L^{-1}$ Fe(II) was measured. However, after the amendment of soil organic matter (SOM) extracts as electron shuttle, the concentration of Fe(II) increased to 7.08-10.87 mmol $L^{-1}$. Regarding the rate of microbial Fe(III) reduction, as shown in Figure S3, the control sample without SOM extracts as electron shuttle showed a maximum reduction rate of 0.79±0.31 mmol Fe(II) $d^{-1}$. The rate of microbial Fe(III) reduction significantly increased with the addition of soil organic matter extracts as electron shuttle, ranging from 0.82-3.05 mmol Fe(II) $d^{-1}$. All of these results are shown in **section 3.3, page 9** of the manuscript.

Regarding the NMR data, I agree that the spectra should be shown in the main paper and a table added to the SI.

According to the editor's suggestion, we added all of the NMR spectra to the main manuscript as Figure 2 and a table showing the relative intensity distributions of specific chemical shifts is shown in the supporting information (Table S2). We also removed the row that showed the aromaticity of samples calculated from the NRM results from Table 1 and combined the aromaticity and the NMR spectra together in Figure 2.

The introduction is missing a lead to the microbial reduction experiment (the abstract too).

To introduce the microbial Fe(III) reduction experiment in the abstract, we added one sentence as *"we also performed a microbial Fe(III) reduction experiment with all of the different extracts and we found that higher EEC..."* **(line 32, page 1)**.

For the introduction, we revised the manuscript from *"We analyzed the electron accepting .... and the electron exchange capacity (the sum of EAC and EDC) of all extracted water-extractable OM, FA and HA fractions and performed a microbial Fe(III) mineral reduction experiment with all of the different extracts"* to *"We analyzed the electron accepting capacity .... and the electron exchange capacity (the sum of EAC and EDC) of all extracted water-extractable OM, FA and HA fractions. To further compare their electron shuttling capacity, we performed a microbial Fe(III) mineral reduction experiment with all of the different extracts"* **(line 85, page 3)**.

The logic flow leading to the microbial Fe(III) reduction in the introduction is, first, we introduced soil organic matter (SOM) and how scientists usually extract humic substance (HS) and use HS as a proxy for SOM (first paragraph). We then explain that participation in redox-active reactions is a very important property of SOM, for example, SOM as electron shuttle to stimulate microbial Fe(III) reduction. However, scientists have been using HS in electron shuttling studies to represent SOM, and since we do not know the impact of the extraction procedure on the redox activities of HS, we could not determine to which extent the electron shuttling capacity of HS is just coming from the extraction-induced changes on the functional groups thus the redox activity (paragraph 2). In paragraph 3 we refer to a previous study that clearly demonstrated higher reducing capacity of the chemically-extracted humic acid compared to water-extracted organic matter. However, in this study the humic acids and the water-extracted organic matter were extracted from different environments. Therefore, in the last paragraph, we explain that we would like to perform an experiment in which we extract SOM with either the traditional chemical method or with water at neutral pH, compare its redox activity by the electron exchange capacity analysis and further test its ability to act as electron shuttle in a microbial Fe(III) reduction experiment.

Include methods for extract chemical analysis.

Thank you for the suggestion. We assume the editor means the methods for the MP-AES analysis of the metal content of our extracts. The results of this analysis are shown in Table S5. Following the reviewer's suggestion, we added section 2.5 in the materials and methods part as "Microwave Plasma-Atomic Emission Spectrometer (MP-AES) analysis" (**Line 136-142, page 5**).

Suggest replacing "setups" with "experiments" or "treatments" throughout.

Thank you for the suggestion. We replaced all "setups" by "experiments" in the manuscript and the supporting information.

Line 12: serving as an electron shuttle, add "an".

Revised, thank you.

Line 18: "led to" instead of "lead to".

As suggested by reviewer 1, this sentence was removed from the abstract. But we carefully went through the manuscript and revised all "lead" to "led" where past tense is needed.

Line 55: Implication, need for research.

We added one sentence here as "*Therefore, studies that compare the stimulating effects of SOM extracted with either the traditional chemical extraction method or with water at neutral pH conditions on microbial Fe(III) reduction are needed*" (**line 65, page 3**).

Line 82: "world reference base for soil*"*, is this correct? It's not capitalized?

Thank you. We revised it to "WORLD REFERENCE BASE FOR SOIL" (**line 94, page 4**).

Line 88: "5000 rpm", can you provide g-force?

Thank you for the suggestion. The rotor we used for centrifugation has a radius of 120 mm, therefore the converted g-force should be 3528×g (added to **line 100, page 4**).

Line 92: Change "doubly-deionized water" to "ultrapure water".

Thank you, changed (**line 103, page 4**).

Line 97: "Filtration", capitalized "F".

Revised, thank you (**line 111, page 4**).

Line 136: samples were p though?

Revised to "*samples were filtered through*" (**line 146, page 5**).

Line 166: "14600 rpm", please give g-force.

Thank you. Revised to 28649 ×g (**line 179, page 6**).

Line 168: Describe AQDS treatment in the section 2.7.

We added "*AQDS, i.e. 2,6-anthraquinone disulphonate, a quinone model compound commonly used in electron shuttling studies that can significantly increase the rates of microbial Fe(III) reduction, was used as 100 μmol L-1 in our experiments as a reference for significant stimulation of Fe(III) reduction by our extracted OM via electron shuttling*" (**line 174, page 6**).

Line 197: "more than 90% of the Fe was removed by filtration", how was this tested?

We first analyzed the concentration of Fe(II) in the water-extractable OM (anoxic) solution right upon finishing the water extraction, and it was 3 mM. When preparing these extracts for the chemical analyses, electron exchange capacity (EEC) analysis, and the microbial Fe(III) reduction experiments, solutions of the extracts were passed through 0.22 μm syringe filters. After this filtration step, we measured again the Fe(II) and Fe(total) concentration in all samples. Taking water-extractable OM under anoxic conditions as an example, the total Fe concentration after filtration was 123.1 μmol L$^{-1}$. This was also the highest Fe concentration we detected among all of our extracts after filtration (Table S3). We calculated that this represents 123.1 μmol L$^{-1}$/3 mmol L$^{-1}$ = 4% of the initial Fe

concentration. This means that the remaining Fe concentration in the water-extractable OM under anoxic condition is up to 4% of the Fe(II) that was detected right after the water extraction, therefore, more than 90% of Fe was removed during the filtration. We used total Fe concentrations instead of Fe(II) concentrations here because all of the extracts, before being prepared into solutions for any analysis or microbial Fe(III) reduction experiment, were freeze-dried under oxic conditions. Therefore, a large portion of Fe(II) was oxidized to Fe(III), and later during the preparation of extracts solution, the pH was adjusted to 7, therefore Fe(III) was not dissolved and got removed by the syringe filter.

In order to show how we got to the conclusion that more than 90% of Fe was removed by filtration, we revised the sentence (line 197, page 7) from *"Although more than 90% of the Fe was removed by filtration (0.22 μm, ...) and around 30% of the remaining Fe(II) was oxidized to Fe(III)...in the water-extractable OM, FA and HA fractions (Tab. S3)."* to *"Although, as shown in Tab. S3, more than 90% of the Fe was removed by filtration (0.22 μm, ...) and around 30% of the remaining Fe(II) ... in the water-extractable OM and FA/HA isolated from it"* (**line 208, page 7**).

Line 215: "The EEC of chemically-extracted FA under anoxic conditions was even 33 μmol e- mmol $C^{-1}$ higher than FA chemically-extracted under oxic conditions". Was this observed in everything with anoxic preparation?

Yes, higher EEC values were observed in all extracts extracted under anoxic conditions. As also shown in section 3.2 in the manuscript**, line 225-234, page 8**, *"The EEC of water-extractable OM obtained under oxic conditions was 32 μmol e- mmol $C^{-1}$ (with ca. 4 μmol e- mmol $C^{-1}$ from Fe), whereas when extracted anoxically, it increased to 44 μmol e- mmol $C^{-1}$ (with 14.8 μmol e- mmol $C^{-1}$ from Fe). Higher EEC values under anoxic compared to oxic extraction conditions were also observed for all extracted FA: for FA isolated oxically from the water-extractable OM, the EEC was 13 μmol e- mmol $C^{-1}$ (2.3 μmol e- mmol $C^{-1}$ from Fe), while it increased to 24 μmol e- mmol $C^{-1}$ (2.7 μmol e- mmol $C^{-1}$ from Fe) when FA was isolated anoxically from the water-extractable OM. The EEC of FA isolated from soil under anoxic conditions was 33 μmol e- mmol $C^{-1}$ higher than FA isolated from soil under oxic conditions. Similar to FA, for the HA isolated from water-extractable OM, the EEC values*

*increased from 15 μmol e- mmol C[-1] (1.9 μmol e- mmol C[-1] from Fe) under oxic conditions to 83 μmol*

*e- mmol C[-1] (7.3 μmol e- mmol C[-1] from Fe) under anoxic conditions. For HA isolated from soil, EEC*

*values increased from 40 μmol e- mmol C[-1] under oxic conditions to 127 μmol e- mmol C[-1] under*

*anoxic conditions".*

Line 228: "strongly".

> Thank you, revised (**line 242, page 8**).

Line 282: "The presence of Fe(II) at the end of extraction in all anoxic extracts suggested that the higher amount of extracted OM is probably related to microbial Fe(III) mineral reduction and the release of mineral-bound OM during mineral dissolution". This is only speculation since you don't show data of Fe-reduction during the extraction.

> First of all, we agree to the editor that this is a speculation. The reason that we do not have a figure or table to show the Fe reduction results during the anoxic water extraction is because we only had the Fe(II) concentration before and after the anoxic extraction, therefore only two data points. This data is shown.

> We speculate that microbial Fe(III) reduction happened during the anoxic water extraction because we measured 3 mM Fe(II) after the anoxic water extraction compared to non-detectable Fe(II) concentrations at the beginning of the anoxic water extraction. Secondly, the soil we used for organic matter extraction was not sterilized, therefore, based on many previous studies, microbial Fe(III) reduction could happen under anoxic and neutral pH conditions. Thirdly, we also observed much more organic carbon extracted by water under anoxic compared to oxic conditions. We assume, according to previous studies (Burdige et al., 1992; Burdige et al., 1999; Chin et al., 1998; Dadi et al., 2017; Gu et al., 1994; Riedel et al., 2013; Shimizu et al., 2013), this is due to the occurrence of microbial Fe(III) reduction under anoxic condition and the release of organic carbon that was co-precipitated with or adsorbed to the Fe(III) minerals.

Line 325-331: Aren't ESI-FTICR-MS the same as FTICR-MS?

ESI, electrospray ionization, is a low-fragmentation ionization technique that preferentially ionizes polar functional groups prior to mass spectrometric analysis. Coupling it to FT-ICR-MS is especially useful for the characterization of organic matter of polar or slightly polar constituents of environmental mixture. Therefore, they are not exactly the same.

Line 342: there are instead of there is.

The discussion part from **line 326-343** was deleted according to the second reviewer's suggestions.

Line 352: add "produced" after mineral phase.

Thank you. Added (**line 348, page 12**).

Table 1: "**The percentage of carbon extracted from soil of the samples are significantly different (n=2, two-sided t-test, P<0.05", the other half bracket missing.

Thank you. Added.

Figure 1: "Forest soil samples (Schönbuch forest, Baden-Wuerttemberg, Germany) dried under 30 ˚C and grinned to pass 2 mm sieve". (1) add 'were' before dried, (2) "grounded" instead of "grinned".

Thank you. The sentence is changed to "*Schematic diagram of soil organic matter (SOM) extraction process. Forest soil samples (Schönbuch forest, Baden-Wuerttemberg, Germany) were dried under 30 ˚C and grounded to pass 2 mm sieve*".

Figure 1: "after centrifugation, the supernatant was acidified by HCl to pH<2, HA then precipitated to be separated from FA". Revise to "HA was then precipitated out to be separated from FA".

Thank you. Revised.

Figure 1: Replace "doubly-deionized water" by "ultrapure water".

Thank you. Revised.

Figure 1: Last sentence in the caption, "This extraction was conducted under both oxic and anoxic conditions", not clear, only the water extraction? Also consider defining abbreviations in the legend.

*We revised the last sentence in the caption to "All of the extractions were conducted under both oxic and anoxic conditions".*

*We also added definitions of all the extracts, as "the supernatant was acidified by HCl to pH<2, HA was then precipitated to be separated from FA, we define these two extracts as HA (isolated from soil) and FA (solated from soil)", "100 g soil was incubated with 400 mL ultrapure water (<18.2 MΩ.cm; Milli-Q, Millipore) at pH 7. pH was monitored during the extraction and it remained stable (range between 7.02-7.33). The slurry was centrifuged after 24 h; we define the supernatant as water-extractable OM" and "the pH of the solution was adjusted to <2 by 6 M HCl to precipitate HA and to separate FA, and these two fractions are HA (isolated from water-extractable OM) and HA (isolated from water-extractable OM)".*

Figure 2: Are these colors color-blind safe? Red-green is a known issue.

*We agree with the editor that red and green are not good colors to present our data. Therefore, we changed FA in Figure2, Figure 3, Figure 4, Figure S2 and Figure S3 into orange and HA in Figure2, Figure 3, Figure 4, Figure S2 and Figure S3 into purple.*

Figure 3: "The insert in penals a-f show the data points for the first 6 h of incubation", revise "6h" to "2 days".

*Revised, thank you.*

Figure 3: Consider using the same scale for y-axis (not for inserts).

*Revised to same scale for all panels, thank you.*

**Reviewer 1 :**

Bai et al. compare redox properties of a water extract with humic and fulvic acid extracts of a soil.

Additionally, fulvic acids and humic acids were extracted of the water extract. All extractions were performed under air as well as under $N_2$. The final conclusion is that (oxic or anoxic) alkaline extracts are not suitable to substitute natural organic matter in redox-experiments. Although the concept of humic and fulvic extractions was often challenged, the method is still popular in current papers and textbooks, sometimes even described as the method of choice. The topic is therefore both important and in time. Also, I appreciate such a clear statement from groups which frequently used humic acids in the past. The paper is well written and nicely illustrated by figures and tables.

One general comment: In the manuscript the abbreviation "SOM" is used to address the water extractable fraction (tables, figures, text). This seems misleading to me: SOM comprises all non-living soil organic matter, while a water extract is done to gain something similar to the dissolved organic matter or the fraction which can be dissolved or mobilized (colloidal fraction) during rain events or rising groundwater. I think, a better name for the water extract would be "water extract (WE)" or "water extractable organic matter (WEOM)" instead of SOM. WE or WEOM also shows that one is aware of the difference between a water extract and the real DOM of a soil solution.

We would like to thank the reviewer for taking the time to go through our manuscript and his/her useful comments, which helped improving our manuscript. We went through the manuscript thoroughly and made sure to carefully incorporate all suggestions and comments.

Following the reviewer's suggestion, we changed the name of this extraction fraction from "water-extracted soil organic matter (SOM)" to "water-extractable organic matter (OM)" in the text, tables and figures in both the manuscript and the supporting information. As suggested by the reviewer, the term "soil organic matter" is usually used to indicate all of the non-living organic matter in soil, which is obviously not possible to be completely extracted by the water extraction method we applied. Instead, our results as presented in Table 1 showed that only about 0.41 % (under oxic conditions) and 2.74 % (under anoxic conditions) of the total organic carbon in the soil was extracted by the water extraction at pH 7. Therefore, in line with the comment provided by the reviewer, we believe that this "water-extracted OM" fraction is the fraction that can be easily mobilized by rain or rising ground water, which is only a small portion of the total soil organic matter (SOM).

To my knowledge, humic substances are meant to extract all HA and FA of a sample, while it is accepted that a large fraction may remain behind. I therefore wonder if it is meaningful to compare extracted C amounts from alkaline extraction with water extraction. With respect to the different extraction of functional groups or artefacts during exposure to pH12, the comparison between WEOM and FA/HA of WEOM are certainly more robust. (For example, does the sum of EAC (FA of WEOM, oxic) and EAC (HA of WEOM oxic) equals EAC(WEOM) or do we see artefacts? The sum of FA of WEOM+HA of WEOM (0.035 g C) is very close to the total WEOM (0.036 g C)).

We agree that the two extraction methods (water vs. alkali/acid) target different fractions of the soil. Actually, in the manuscript, we pointed out that chemical extraction at pH 12 leads to the extraction of different fractions of soil organic material compared to water extraction. In **line 322**, **page 11**, we wrote "However, we found that even under anoxic conditions, the chemical extraction extracted up to 100 times more carbon than the water extraction at neutral pH (Tab. 1), consistent with previous studies (Aiken, 1985). This higher extraction efficiency at high pH could be due to the deprotonation of carboxyl and phenol functional groups leading to both higher aqueous solubility and electrostatic repulsion of OM from negatively charged soil minerals (Kleber et al., 2015) or due to the hydrolysis of plant material and the formation of smaller oligo- and monomers (Sparks, 2003)." Nevertheless, we agree with the reviewer that comparing the chemical characteristics and redox properties of the FA and HA chemically isolated from the water-extractable OM (the reviewer calls it „FA/HA of WEOM") to the water-extractable OM itself is more straightforward and informative than comparing the chemically extracted FA and HA from soil to water-extractable OM. Therefore, we will revise our manuscript and when presenting the data regarding the amount of carbon extracted, the SUVA$_{254}$ values, the NMR, emission-excitation matrix (EEM) and electron exchange capacity (EEC) results, we will compare and discuss the results between FA and HA isolated from the water-extractable OM to the water-extractable OM. We would like to make revisions in the manuscript as follows:

**Line 190, page 7**, change from

"*Chemical extraction using NaOH and HCl under oxic conditions yielded ca. 100 times more carbon*

*for FA (1.451 g; 17.0% of the total carbon present in the soil) and HA (1.450 g; 17.0%) compared to water under oxic conditions at neutral pH. Under anoxic conditions, the chemical extraction also lead to a somewhat higher percentage of extracted carbon for FA (20.7%) and HA (22.0%) than under oxic conditions",*

to

*"Chemical extraction directly from the soil using NaOH and HCl under oxic conditions yielded 1.451 g C in the extracted FA (17.0% of the total carbon present in the soil) and 1.450 g C in HA (17.0% of the total carbon present in the soil). Conducting the same chemical extraction from soil under anoxic condition lead to higher percentage of extracted carbon for FA (20.7%) and HA (22.0%) than under oxic conditions".*

**Line 197, page 7**, change from

*"All HA extracts showed much higher $SUVA_{254}$ values than SOM and FA extracts, with the highest value of 0.265 $mg^{-1}$ C $cm^{-1}$ (HA chemically-extracted under anoxic conditions), followed by HA chemically isolated from the water- extracted SOM under oxic conditions (0.207 $mg^{-1}$ C $cm^{-1}$). For HA chemically isolated from the water-extracted SOM under anoxic conditions and HA chemically extracted from soil under oxic conditions, the SUVA254 values are 0.068 and 0.083 $mg^{-1}$ C $cm^{-1}$, respectively."*

to

*"HA extracts isolated from the water-extractable OM under oxic condition showed a $SUVA_{254}$ value of 0.068 $mg^{-1}$ C $cm^{-1}$, higher than the 0.018 $mg^{-1}$ C $cm^{-1}$ of the water-extractable OM itself obtained under oxic conditions. A higher $SUVA_{254}$ value for the HA isolated from the water-extractable OM (0.207 $mg^{-1}$ C $cm^{-1}$) than for the water-extractable OM itself (0.027 $mg^{-1}$ C $cm^{-1}$) was also found under anoxic extraction conditions".*

The reviewer also suggested to compare the electron accepting capacity (EAC) of the FA and HA isolated from the water-extractable OM to that of the water-extractable OM to clarify whether the chemical extraction method modifies the chemical composition and the redox properties of the

extracts. In the manuscript, we therefore compare the "total number of electrons that can be transferred" (the electron exchange capacity) of FA and HA isolated from the water-extractable OM to the values of the water-extractable OM, these results are presented in Figure 2. In **line 235-242, page 8**, we write "*The total number of electrons that can be exchanged (that means transferred from Fe(III)-reducing bacteria to the OM, or from the OM to Fe(III) minerals) by water-extractable OM before and after the chemical separation of FA and HA from this water-extractable OM was also calculated (the recovery of EEC) under both oxic and anoxic conditions (Fig. 2). For the extracts obtained under anoxic conditions, the sum of total exchangeable electrons values of the FA and HA isolated from water-extractable OM (786 mmol e-) was almost identical to that of water-extractable OM itself (836 mmol e-). In contrast, under oxic conditions, the sum of the EEC values of the FA and HA separated from the water-extractable OM was 324 mmol e-, ca. 5-times higher than the EEC value of the water-extractable OM (64 mmol e-). This confirms that the traditional chemical extraction procedure conducted under oxic conditions strongly enhances the redox capacity of the samples*". We compared the "total amount of electrons that can be transferred" (the recovery of electron exchange capacity (EEC)) instead of electron accepting capacity (EAC) as suggested by the reviewer because while the EAC represents mainly the content of quinone functional groups in the extracts, the electron donating capacity (EDC) can also provide information for the abundance of functional groups such as phenols in the extracts (Aeschbacher et al., 2012). Therefore, to understand the effects of the chemical extraction method on the content and abundance of functional groups of the soil extracts, we think that the total EEC should be used instead of EAC only.

It was also pointed out by the reviewer that under oxic extraction conditions, the sum of the carbon content of FA and HA from water-extractable OM (0.021+0.014=0.035 g C) is very close to the C of the total water-extractable OM (0.036 g C). Under anoxic extraction conditions, the carbon content of FA and HA isolated from water-extractable OM was 0.146 and 0,079 g, respectively. And the sum of them (0.225 g) is also very close to the carbon content of water-extractable OM under anoxic conditions (0.234 g). We think this result makes sense because under both oxic and anoxic conditions, the FA and HA are a sub-fraction of the water-extractable OM, therefore the carbon content of FA and HA should each be lower than the carbon content of the water-extractable OM (but

the sum of HA+FA should represent the OM). The fact that the sum of the carbon content of FA and HA is very close to the carbon content of the water-extractable OM indicates that the chemical treatment of the OM water extract with HA and HCl is very effective and complete without any significant C loss.

Other comments:

First sentence: I propose to delete "including humic substances". The main point can be made without and if alkaline extraction causes that many artefacts, it may even be wrong.

As suggested, we would like to delete "including humic substances" and revise the sentence to "*Soil organic matter (SOM) is redox-active, can be microbially reduced, and transfers electrons in an abiotic reaction to Fe(III) minerals thus serving as electron shuttle.*"

Line 15: name soil, sampled depth/horizon and pH of the soil.

The sentence of **line 15** will be revised from "*we prepared HS and SOM extracts from a forest soil applying either a combination of 0.1 M NaOH and 6 M HCl, or water (pH 7)*"

to

"*we prepared HA, FA and water-extractable organic matter (OM) extracts, applying either a combination of 0.1 M NaOH and 6 M HCl or ultrapure water (pH 7), from soil samples collected from the subsoil (0-15 cm, A horizon, pH 6.5-6.8) in Schönbuch forest, Baden-Wuerttemberg, Germany.*" (**line 15, page 1**).

Line 18/19: "…100 times more…" see general comment above.

As suggested by the reviewer, we would like to revise the sentence from "*We found that soil extraction with NaOH lead to ca. 100 times more extracted C and the extracted HS had 2-3 times higher electron exchange capacities (EEC) than SOM extracted by water.*"

to

"*We found that FA and HA chemically extracted from the soil can make up to 34-40% of the soil*

*organic carbon pool while the water-extractable OM only represents 0.41-2.74% of the total soil organic carbon. The higher extraction efficiency of the chemical extraction is probably due to the deprotonation of carboxyl and phenol functional under high pH*" (**line 21, page 1**).

Line 25: . . .changes in functional groups . . . is there space to name observed or assumed changes in the abstract?

To include the reviewer's suggestion, we would like to revise the text from "*NaOH/HCl treatment of the water-extracted SOM lead to 2 times higher EEC values in the HA isolated from the SOM compared to the water-extracted SOM itself, suggesting the chemical treatment with NaOH and HCl caused changes of redox-active functional groups of the extracted organic compounds.*"

to

"*under anoxic extraction condition, the HA chemically isolated from the water-extractable OM had 2 times higher EEC values compare to the water-extractable OM itself, suggesting the potential formation of redox-active aromatic functional groups during the extraction with NaOH under anoxic conditions by condensation reactions between amino acids, aldehydes, hydroxyl-and catechol-containing molecules.*" (**line 29, page 1**).

Line 27: Delete "at neutral pH". Rainwater has a pH of 5, soil solutions are rather variable in pH. The choice of pH should therefore be adopted to the soil and the specific research question. Ionic strength and ionic composition will also play a role. Was pH7 a good choice for the Cambisol?

We agree with the reviewer that rainwater has a pH of about 5.6 (Liljestrand, 1985), and the soil solution pH of Cambisol soils ranges between 5.8-6.2 (WRB). However, we chose pH 7 for the extraction because, first of all, in a paper published in 2014 by our group, the pH of the soil from exactly the same sampling site in the Schönbuch forest was measured 24 h after the addition of 0.01 M $CaCl_2$, and was reported as $7.0\pm0.0$ (Porsch et al., 2014). In our experiment, the pH of the native soil slurry in Milli-Q water ranged from 6.5-6.8. Therefore, we think that pH 7 as the extraction pH is relevant to the native soil pH conditions. Additionally, we also believe that the extraction pH has significant impact on the results from the electrochemical analysis and the microbial experiments

(Aeschbacher et al., 2012). The standard pH for the electrochemical analysis is 7 (Aeschbacher et al., 2012; Aeschbacher et al., 2010) and for the microbial Fe(III) reduction experiments, also pH 7 is used because pH 7 is the optimal growth pH of *Shewanella oneidensis* MR-1 (Babauta et al., 2011). If we conduct water extraction of OM under the native pH of soil at around 6.5-6.8, and later adjust the pH of the water-extractable OM to 7 for the analyses, due to the possible alteration of the functional groups of the OM by changing of pH and by the use of NaOH for the pH adjustment (Dadi et al., 2017; Guigue et al., 2014; Kappler and Brune, 1999; Schnitzer and Skinner, 1968), it is difficult to judge whether and to which extent the electrochemical capacities and the extent and rate of microbial Fe(III) reduction are a result of the artifacts caused by the pH-adjusting step. We would therefore like to add one sentence to explain this in section 2.1 (**line 104, page 4**) as follows: "*The pH of the water extraction was adjusted to 7 to avoid any possible artefacts resulting from further pH adjustments prior to analyses and experiments that require the sample pH to be 7 such as the electrochemical analysis or the microbial Fe(III) reduction*".

However, we are aware that adjusting the pH using 1 M NaOH from pH 6.5-6.8 to pH 7 will increase the ionic strength of the solution used for the extraction. To adjust 400 mL of soil slurry solution to pH 7, we used maximally 0.5 mL 1 M NaOH. Therefore, the final concentration of NaOH in the soil slurry is:

$1 \text{ mol L}^{-1} \times 0.5 \times 10^{-3} \text{ L} = 0.5 \times 10^{-3} \text{ mol} \div 400 \times 10^{-3} \text{ L} = 0.00175 \text{ mol L}^{-1}$

The ionic strength of 0.00175 mol L-1 NaOH is:

$I = \frac{1}{2} \{0.00175 \times (+1)^2 + 0.00175 \times (-1)^2\} = 0.00175 \text{ mol L}^{-1}$

The measured ionic strength of the soil solution of Schönbuch forest is $0.005 \pm 0.003 \text{ mol L}^{-1}$ (Rothe et al., 2002), and therefore an increase by $0.00175 \text{ mol L}^{-1}$ from the NaOH used to adjust the pH is smaller than the ionic strength stemming of the soil. Furthermore, although it is known that the amount of OM extracted decreases with increasing ionic strength, significant changes in the amount of OM that was extracted was only seen when the increase of the ionic strength was at least one order of magnitude (Evans and Sorensen, 1985). Therefore, we think in our system, increasing the ionic strength by $0.00175 \text{ mol L}^{-1}$ on top of the $0.005 \text{ mol L}^{-1}$ stemming from the native soil ionic strength should not have a significant impact on the amount of carbon that can be extracted from the soil.

Line 81: Was the mineral topsoil horizon sampled or the mineral topsoil plus litter layer? Do the upper 15 cm include material of the B horizon? Is anything known about the Fe oxide content of the soil? If yes, does the amount of Fe oxides fit to the observed Fe concentration in the extracts?

We only sampled the topsoil horizon without any of the litter layer. The soil sample 0-15 cm was classified as A horizon according to the German soil classification (Classification, 1997; Weigold et al., 2016) therefore we believe it does not include any material from the B horizon. Consequently, the sentence in **line 92, page 3**, "*Top soil (0-15 cm) was collected from the Schönbuch forest, Baden-Wuerttemberg, Germany.*" will be revised to "*Top soil (0-15 cm) without leaf litter from A horizon was collected form Schönbuch forest, Baden-Wuerttemberg, Germany.*"

The 0.5 M-HCl extractable total Fe (poorly-crystallized Fe) mass fraction of the soil is 0.2 wt%. Therefore, in the 100 g soil we used for the extraction, the poorly-crystallized Fe is:

$100 \text{ g} \times 0.2\% = 0.2 \text{ g}$

The molecular weight of Fe is 56 g/mol, therefore, the molar amount of the Fe in 100 g soil is

$0.2 \text{ g} \div 56 \text{ g/mol} = 0.00357 \text{ mol} = 3.57 \text{ mmol}$

The extraction solution was 400 mL, therefore the concentration of Fe in the water-extractable OM solution is:

$3.57 \text{ mmol} \div 400 \text{ mL} = 0.0089 \text{mol L}^{-1} = 8.9 \text{ mmol L}^{-1}$

The total Fe concentration after 1 h of extraction by 1 M HCl was 3 mmol L$^{-1}$, measured right after the water extraction under anoxic conditions. Under oxic extraction conditions, the total Fe concentration decreased to 0.8 mmol L$^{-1}$. The reason for the higher Fe concentration in the extracts under anoxic conditions is the microbial reduction of Fe(III) minerals, as also discussed in the manuscript. As calculated above, both Fe concentrations measured under anoxic and oxic extraction conditions were in the range of the amount of poorly-crystalline Fe present in the soil sample (8.9 mmol L$^{-1}$).

The total Fe concentrations in the water-extractable OM samples extracted under both oxic and anoxic conditions were analyzed again after filtration of the samples through 0.22 μm syringe filters before electrochemical analyses and before addition to the microbial Fe(III) reduction (see

values in Table S3). Because the filtration step removed a large amount of Fe from the extracted OM sample, after filtration the total Fe concentration in the water-extractable OM was 32.18 μmol L$^{-1}$ and 123.1 μmol L$^{-1}$, respectively. These concentrations were also used for the calculation of the Fe contribution to the electron exchange capacity of the extracted OM samples because the OM samples were also filtered through 0.22 μm syringe filters before electrochemical analysis.

Line 63: please give ferrihydrite concentration in mM Fh-Fe or in weight %. As the composition of ferrihydrite is still under debate and as a mineral structure can be given in different ways, mM Fh is not unambiguous.

We would like to thank the reviewer for this suggestion (he/she probably meant line 163). The composition/formula of ferrihydrite is indeed still under debate so giving the concentration as 15 mM ferrihydrite is somehow ambiguous. Therefore, to be more accurate, we would like to replace all "*15 mM ferrihydrite*" in the manuscript by "*15 mM Fe(III)*". The reason is that after we synthesized the ferrihydrite, we analyzed the Fe(III) concentration in the ferrihydrite by ferrozine assay (Stookey, 1970) and for the microbial Fe(III) reduction experiment, the amount of ferrihydrite added was calculated based on the measured Fe(III) concentration of ferrihydrite. Therefore the targeting concentration (15 mM) in the microbial Fe(III) reduction setups represent Fe as Fe(III).

Line 194: delete the "e"

Thank you. Deleted. (**line 205, page 7**).

Line 211: please correct, the table in Fig.2 give 44 μmol e- mmolC$^{-1}$, instead of 45.

Corrected. Thank you. (**Line 226, page 8**).

Line 304-307: sentence?

We split the long sentence, "*Consistent with previous studies, our SUVA$_{254}$ (Tab. 1), EEM (Fig. S1), $^{13}$C-NMR (Tab. S2) and EEC (Fig. 2) results showed lower aromaticity and EEC of chemically-extracted FA and HA under oxic conditions compared to chemically-extracted FA and HA*

*extracted under anoxic conditions, indicating the degradation of aromatic structures and functional groups in the OM to smaller molecules in the presence of $O_2$ under high pH conditions"* into two sentences: *"Consistent with previous studies, our $SUVA_{254}$ (Tab. 1), $^{13}C$-NMR (Tab. S2) and EEC (Fig. 2) results showed that, FA and HA isolated from soil under oxic conditions had lower aromaticity and EEC compare to the FA and HA isolated from soil under anoxic condition. This indicates that degradation of aromatic structures and functional groups in the OM to smaller molecules occurs in the presence of $O_2$ under higher pH conditions"* (**line 317, page 11**).

Line 400: "adsorption" instead of "absorption"

Revised. Thank you (**Line 398, page 13**).

Table 1 is unclear. What was analyzed, DOC (after 0.45 filtration) or TOC? Why are concentrations and masses of C needed? Can the descriptive first and second rows of the table be optimized? "Water extracted" above columns 4,5,8, and 9 is misleading, as the treatment is a chemical extraction of a water extract. The latter also applies for the figures. Replace SOM by WE or WEOM or the like.

First, we agree with the reviewer that naming the FA and HA isolated from the water-extracted OM as "*water-extracted HA/FA*" is misleading. Therefore, we would like to name the FA (and the HA) isolated from the water-extractable OM as "*FA (or HA) isolated from water extract*", whereas the FA (or HA) extracted chemically from soil directly as "*FA (or HA) isolated from soil*". The terminology for all extracts will be revised in the method section 2.1 (**pages 3-4**) and throughout the manuscript, the supporting information and in all tables and figures.

Second, regarding the question whether dissolved organic carbon (DOC) or total organic carbon (TOC) of the extracts was analyzed, we analyzed DOC directly from the extracted solution (after passing through 0.45 mm syringe filter) for the water-extractable OM (oxic or anoxic), and for the FA iosolated from soil (oxic or anoxic). For the FA isolated from the water-extractable OM (oxic, anoxic) and of all the precipitated HA samples, the TOC was analyzed from the freeze-dried samples. This is explained in the first sentence below Table 1. However, to make it clearer, we will revise this sentence to "*TOC of all HA extracts and of the FA isolated from the water-extracted OM was directly*

*quantified from the freeze-dried powders. The carbon content of all other liquid samples was determined as DOC, measured directly from the solutions after passing through 0.45 mm syringe filter".*

In the second row, we show the "*total organic carbon of the extracts*", that means the total mass of carbon that was extracted. This value was calculated by multiplying the measured OC concentration of the extracted solutions with the volume of the extracted solution. In order to make it clearer, and as suggested by the reviewer "to optimize the first two rows", we revised the second sentence below Table 1 to "*total organic carbon content in the extract was directly quantified form the freeze-dried samples of FA isolated from the water-extractable OM (oxic, anoxic) and of the HA extracts". For the other extract solutions, the total organic carbon was calculated by DOC (mg C L$^{-1}$) ×volume of the extracted solution (L).*"

The revised Table 1 is shown below, all revisions are shown in blue.

| | Water-extractable OM | | FA | | | | [a]HA | | | |
|---|---|---|---|---|---|---|---|---|---|---|
| | Water-extracted, oxic | Water-extracted, anoxic** | [a]Isolated from water-extractable OM, oxic | [a]Isolated from water-extractable OM, anoxic** | Isolated from soil, oxic | Isolated from soil, anoxic** | Isolated from water-extractable OM, oxic | Isolated from water-extractable OM, anoxic** | Isolated from soil, oxic | Isolated from soil, anoxic** |
| DOC concentration in extract (mg C L$^{-1}$) | 0.149 ± 0.036 | 0.890 ± 0.041 | – | – | 5.800 ± 0.025 | 6.320 ± 0.071 | – | – | – | – |
| [b]Total organic carbon in extract (g) | 0.036 ± 0.012 | 0.234 ± 0.015 | 0.021 ± 0.002 | 0.146 ± 0.013 | 1.451 ± 0.008 | 1.770 ± 0.028 | 0.014 ± 0.003 | 0.079 ± 0.000 | 1.450 ± 0.002 | 1.881 ± 0.029 |
| [c]Percentage of carbon extracted from soil (%) | 0.41 ± 0.14 | 2.74 ± 0.18 | 0.24 ± 0.02 | 1.69 ± 0.15 | 17.0 ± 0.09 | 20.7 ± 0.32 | 0.15 ± 0.03 | 0.90 ± 0.00 | 17.0 ± 0.02 | 22.0 ± 0.34 |
| [d]SUVA$_{254}$ (mg$^{-1}$ C cm$^{-1}$) | 0.018 | 0.027 | 0.017 | 0.029 | 0.023 | 0.042 | 0.068 | 0.207 | 0.083 | 0.265 |

[a]TOC of all HA extracts and of the FA isolated from the water-extractable OM was directly quantified from the freeze-dried powders. The carbon content of all other liquid samples was determined as DOC, measured directly from the solutions after passing through 0.45 μm syringe filter

[b]Total organic carbon content in the extract was directly quantified form the freeze-dried samples of FA isolated from the water-extractable OM (oxic, anoxic) and of the HA extracts". For the other extract solutions, the total organic carbon was calculated by DOC (mg C L-1) $\times$ volume of the extracted solution (L).

[c]Percentage of carbon extracted from soil = Total carbon extracted (mg g$^{-1}$)/soil carbon content (8.54 mg g$^{-1}$)

[d]Specific UV absorbance 254 nm (SUVA $_{254}$) = UV254 $\times$ DOC (mg C L$^{-1}$), b is the optical path length in centimeter (1 cm in this experiment)

**The percentage of carbon extracted from soil of the samples are significantly different (n=2, two-sided t-test, $P < 0.05$)

Figure 3 is very confusing due to the complex legend.

The revised Figure 3 according to the reviewer's suggestion is shown below. We added a legend for each single figure right on top or beneath that figure.

[Figure]

Figure 1, although nice, could be sacrificed to show NMR or EEMS data. It's a pity, that these ended up in the SI. Can NMR spectra be given at least in the SI?

We would like to thank the reviewer for this suggestion. However, if possible (if the reviewer and editor agree) we would prefer to keep Figure 1 due to the following reasons: we believe that this figure clearly shows the extraction process, so even readers who will not read the methods section in

detail will get an idea about what all of the extracts are and how were they extracted. We decided to put the EEM spectra in the SI because there have been debates about the suitability of EEMs, specifically whether EEM analysis reflects the redox state and thus the abundance of redox-active functional groups and aromaticity of humic substances (HS) and natural organic matter (NOM). For example, Maurer et al. (Maurer et al., 2010) compared the EEM spectra of native, electrochemically reduced and re-oxidized humic acid (HA) and they did not find any difference in the emission peak position and density. Identical EEM spectra between native and electrochemically reduced lake organic matter were also shown in the study of Fimmen et al. (Fimmen et al., 2007). Therefore, instead of putting too much emphasis on the EEMs, we use $SUVA_{254}$ (results shown in Figure 1), specific UV absorbance at 254 nm, to characterize the aromaticity of our extracts. $SUVA_{254}$ is a well-recognized method and it has been shown to be a very useful proxy for OM aromatic content (Hansen et al., 2016; Weishaar et al., 2003) and molecular weight (Chowdhury, 2013).

Regarding NMR: as suggested by the reviewer, we would like to show all the NMR spectra in the manuscript (Figure 2) and show the calculated aromaticity of the extracts from the NMR spectra (as suggested by the study of Abelmann et al. (Abelmann et al., 2005)) by adding an extra table in Figure 2.

Title: While the manuscript is concise and clear, the title is less straight. How about having the main message in the title?

As suggested by the reviewer, we would like to revise the title to "High pH and anoxic conditions during soil organic matter extraction increase its electron exchange capacity and its ability to stimulate microbial Fe(III) mineral reduction by electron shuttling".

Reviewer 2:

This paper compares how different extraction methods influence the composition of soil organic matter (SOM) derived from the process. They compared SOM extracted by neutral pH water and mediated by alkaline extraction followed by acid precipitation (the standard approach used to delineate soil humic and fulvic acids) under oxic and anoxic conditions. The authors determined carbon recovered, specific UV absorbance @ 254 nm (SUVA), and most importantly the exchangeable electron capacity (EEC) as well as electron accepting and donating capacities (EAC and EDC). The manuscript is well-organized, and easy to read (even though there are a couple of typos that spell checker did not catch). The most important contribution, however, is the electrochemical analyses that were conducted, which makes this paper really unique. There are a few major issues that I have, and specific comments are below.

We would like to thank the reviewer for going through our manuscript and providing useful suggestions to help us to improve the manuscript. We also appreciate the overall positive response and in particular his/her appreciation of the electrochemical analyses of the SOM extracts. We will go through the manuscript carefully and incorporate all suggestions and comments.

1. The SUVA data seems fine for the water extracted SOM and fall well within the range of values reported by others (e.g., Weishaar et al., 2003). However, the FA and HA alkaline extraction conducted anoxically were off the charts and many factors higher than the highest value reported by Weishaar et al., 2003. These numbers appear unrealistic and could be due to the presence of iron (both (II) and (III)) in the extracts that reached 3 mM. Given that Weishaar et al., reported iron interference (they use Fe(III) as an example, but noted that Fe(II) can also interfere) at levels of only a few mg/L (or 10's of μM) this could be a positive interference to their SUVA data.

We agree with the reviewer that the $SUVA_{254}$ values measured in our study, especially for the HA isolated from the water-extracted OM (0.207 L $mg^{-1}$ C $cm^{-1}$), are almost one order of magnitude higher than the reported $SUVA_{254}$ values for HA chemically extracted from Coal Creek

soil in Weishaar et al., 2003 (0.039 mg$^{-1}$ C cm$^{-1}$) (Weishaar et al., 2003), and also higher than the typical SUVA$_{254}$ values of HA analyzed in many other studies (Beckett et al., 1987; Chen et al., 2003; Fox et al., 2017). However, the SUVA$_{254}$ of all our FA extracts range from 0.017 to 0.042 L mg$^{-1}$ C cm$^{-1}$, and these values are in line with previous studies (Beckett et al., 1987; Chen et al., 2003; Fox et al., 2017).

One reason for the higher SUVA$_{254}$ values for the HA isolated from water-extracted SOM under anoxic conditions in our study compared to others, in addition to the differences in the soils from which the HA were extracted, as suggested by the reviewer, could be the presence of Fe(II) and Fe(III). As shown by Weishaar et al. (2003), the presence of 4 mg L$^{-1}$ Fe(III) showed an absorbance value about 0.65 cm$^{-1}$ at 254 nm wavelength, and the absorbance increased with increasing Fe(III) concentrations. Based on this study, we can hypothesize that also in our case the presence of Fe(III) influenced the measured SUVA$_{254}$ value of the HA isolated from the water-extracted SOM under anoxic conditions. The Fe(III) concentration in the HA isolated from water-extracted SOM under anoxic condition was 33 μmol L$^{-1}$. Please note that we did not use the concentration of 3 mmol L$^{-1}$ as suggested by the reviewer, because 3 mmol L$^{-1}$ was the Fe(II) concentration determined right after the anoxic water extraction of SOM. However, all of the samples were passed through 0.45 mm syringe filters under oxic conditions before the SUVA analyses. Therefore, a large amount of the Fe(II) was oxidized to Fe(III) and removed as particulate Fe(III) by the filtration, as explained in the manuscript **line 208-211, page 7**. The remaining Fe(II) and Fe(III) in the samples were analyzed and shown in Table S3 and the Fe(III) concentration for HA isolated from the water-extracted SOM under anoxic condition was 33 μmol L$^{-1}$.

33 μmol L$^{-1}$ × 56 g mol$^{-1}$ = 1.848 mg L$^{-1}$

According to Weishaar et al. (2003), 1.848 mg L$^{-1}$ Fe(III) has an absorbance value of 0.15 cm$^{-1}$ at 254 nm wavelength. With the additional 28 μmol L$^{-1}$ Fe(II), the contribution of iron to the

measured SUVA$_{254}$ value should be even higher. Therefore, we would like to add one sentence in the results section 3.1, **line 212, page 7**, as "*A previous study showed that 4 mg L$^{-1}$ Fe(III) yielded an absorbance value of 0.65 cm$^{-1}$ at 254 nm wavelength (Weishaar et al., 2003). Therefore, we believe that the high SUVA$_{254}$ value of HA isolated from the water-extractable OM (please note, this is the term used to replace 'water-extracted SOM' as suggested by reviewer 1) compared to SUVA$_{254}$ values of HA shown in previous studies could be caused by the presence of Fe(II) and Fe(III) in the sample due to the microbial Fe(III) reduction that occurred under the anoxic extraction conditions.*"

2. There was only passing mention of the NMR and fluorescence data. Why wasn't this data more prominently discussed in the paper (as opposed to a glancing mention in the SI)? For example, how does the EEM data "confirm higher contents of aromatic carbon" (the ex- planation in the SI caption was inadequate)? Further, the relatively smaller differences in NMR determined aromaticity between anoxically extracted vs oxic extraction SOM is not reflected in the much larger (order of magnitude) spread observed for SUVA (see above). Further, the EEMs from Figure S2 look really odd and I suspect that this caused by the really high DOC levels used by the authors (100 mg/L!). At those levels inner- filter-effects will become dominant as the solution will be optically dense to the point where inner-filter corrections will likely no longer work. Typically, fluorescence EEMs are collected at much lower (nearly two orders of magnitude) DOC concentrations to minimize inner-filter-effects (see papers by Stedmon et al., in L and O). Thus, because the data is likely improperly collected I would simply eliminate it from the discussion.

We would like to thank the reviewer for the comments and suggestions. First, we would like to clarify that the concentration of all samples used for the EEM analysis was not 100 mg C L$^{-1}$ In the manuscript, **line 113, page 4**, it says "*Freeze-dried SOM/FA/HA powders were dissolved in Milli-Q water (pH 7) at a concentration of 100 mg C L$^{-1}$ and the solutions were agitated for 12 h at 300 rpm at room temperature, samples were then filtered through 0.45 mm syringe filter*

*(mixed cellulose ester (MCE, Millipore, Germany). For fluorescence analyses, samples were prepared by stepwise dilution of extract solution with Milli-Q water (pH 7) until absorbance values of 0.300 at 254 nm wavelength were reached".* Therefore, after the stepwise dilution, the final concentration of samples used for the EEM analyses was much lower than 100 mg C $L^{-1}$ and inner-filter effects can be neglected.

However, as the reviewer pointed out, we did not draw any conclusions directly from the EEM data. As also explained in the reply letter to reviewer 1, we did this because there are many debates about whether, to any extent, the EEM spectra can reflect the redox state and the aromaticity of the OM samples (Fimmen et al., 2007; Maurer et al., 2010). Since only very briefly mentioned the results of the EEM spectra and since leaving the EEMs out completely will not impact the conclusion of the manuscript at all, as suggested by the reviewer, we would like to completely remove the EEM results from our manuscript.

Regarding the NMR, as we commented already in the reply letter to reviewer 1, we would like to show in the revised manuscript the NMR spectra of all samples as Figure 2.

Finally, regarding the reviewer question "why the very distinct difference between different OM extracts as shown by the SUVA results could not be seen in the calculated aromaticity from NMR": in previous studies that applied NMR to characterize the aromaticity of OM, the differences between different OM samples are usually in the range of 10%. For example, in the study of Lorenz et al. (2006), seven different OM samples extracted from different sampling sites were analyzed, and the aromaticity of these samples ranged from 21-32%. Inbar and co-authors compared the aromaticity of native SOM and the same SOM after 147 days of composting, and the aromaticity only changed from 35% to 37% (Inbar et al., 1990). In our study, all of the OM extracts, although extracted in different ways, were from the same soil. Therefore, we believe that, for example, the 4% difference in the aromaticity between chemically-extracted HA under oxic and chemically-extracted HA under anoxic condition does indicate a potential difference in

the aromaticity of these two samples. Moreover, although the differences of the aromaticity among different OM samples calculated from NMR are not as significant as the differences of $SUVA_{254}$ values among different samples, the aromaticity of SOM/HA/FA extracted under anoxic conditions was higher than the aromaticity of SOM/HA/FA extracted under oxic conditions. Furthermore, under anoxic conditions, FA and HA isolated from the water-extracted SOM (water-extractable OM) both have higher aromaticity than the water-extractable OM itself. Therefore, our NMR results are perfectly in line with the $SUVA_{254}$ results and the electron exchange capacity analysis of the OM samples and can be used to support and strength our argument that the chemical extraction and the presence of oxygen impacts the aromaticity thus the redox activity of the SOM extracts.

3.  I think the discussion regarding the comparison between Suwannee River reverse osmosis dissolved organic matter (DOM) to the fulvic acid fraction isolated by XAD-8 chromatography (as opposed to acid precipitation) does not add any value to the paper because you are basically comparing apples and oranges (i.e., SOM vs. aquatic DOM). The methods are totally different from alkaline and neutral extraction and there are no mineral phases involved. The authors can delete the entire discussion and it will not affect the conclusions or the quality of this paper.

    Following the reviewer's suggestion, we would like to remove this part of the discussion: **line 326-343, page 11**.

4.  While the authors point to several studies demonstrating correlations between DOC and Fe(II) formed from the dissolution of iron oxides in batch incubation studies, evidence for this relationship has also been reported in benthic pore waters. See papers bu Burdige (et al.,), Chin (et al.,), plus many others. I think showing that this phenomenon occurs in real aquatic systems strengthens the arguments put forth by the authors for this paper.
    As suggested by the reviewer, to strengthen our argument of the correlation between DOC and Fe(III) mineral dissolution, we would like to add one sentence as follows (at **line 297, page 10**):

*"In-situ monitoring of the DOC flux in pore water of marine sediment or freshwater wetland also*

*suggested an increase in DOC with increasing amount of microbial iron(III) mineral reduction*

*(Burdige et al., 1992; Burdige et al., 1999; Chin et al., 1998)".*

[revised manuscript text omitted]

**Calculation of contribution of Fe(II) to the EDC:**

Take water-extractable OM, oxic as an example, Fe(II) concentration = 17.4 µmol L⁻¹

The volume of OM solution used for EDC analysis is 200 µL

Mole quantity of Fe(II) = 17.4 µmol L⁻¹ * 200 µL = 0.00348 µmol Fe(II)

[revised manuscript text omitted]

Yuge Bai 17.12.2019 10:54

Yuge Bai 17.12.2019 10:53

Yuge Bai 17.12.2019 10:52